# The HTLV-1 viral oncoproteins Tax and HBZ reprogram the cellular mRNA splicing landscape

Charlotte Vandermeulen[1,2,3], Tina O'Grady[3], Jerome Wayet[4], Bartimee Galvan[3], Sibusiso Maseko[1], Majid Cherkaoui[1], Alice Desbuleux[1,2,5,6], Georges Coppin[1,2,5,6], Julien Olivet[1,2,5,6], Lamya Ben Ameur[7], Keisuke Kataoka[8], Seishi Ogawa[8], Olivier Hermine[9], Ambroise Marcais[9], Marc Thiry[10], Franck Mortreux[7], Michael A. Calderwood[2,5,6], Johan Van Weyenbergh[11], Jean-Marie Peloponese[12], Benoit Charloteaux[2,5,6,13], Anne Van den Broeke[4,14]*, David E. Hill[2,5,6]*, Marc Vidal[2,5]*, Franck Dequiedt[3]*, Jean-Claude Twizere[1,2]*

**1** Laboratory of Viral Interactomes, GIGA Institute, University of Liege, Liege, Belgium, **2** Center for Cancer Systems Biology (CCSB), Dana-Farber Cancer Institute, Boston, Massachusetts, United States of America, **3** Laboratory of Gene Expression and Cancer, GIGA Institute, University of Liege, Liege, Belgium, **4** Unit of Animal Genomics, GIGA, Université de Liège (ULiège), Liège, Belgium, **5** Department of Genetics, Blavatnik Institute, Harvard Medical School, Boston, Massachusetts, United States of America, **6** Department of Cancer Biology, Dana-Farber Cancer Institute, Boston, Massachusetts, United States of America, **7** Laboratory of Biology and Modeling of the Cell, CNRS UMR 5239, INSERM U1210, University of Lyon, Lyon, France, **8** Department of Pathology and Tumor Biology, Graduate School of Medicine, Kyoto University, Kyoto, Japan, **9** Service Hématologie Adultes, Assistance Publique-Hôpitaux de Paris, Hôpital Necker Enfants Malades, Université de Paris, Laboratoire d'onco-hématologie, Institut Necker-Enfants Malades, INSERM U1151, Université de Paris, Paris, France, **10** Unit of Cell and Tissue Biology, GIGA Institute, University of Liege, Liege, Belgium, **11** Laboratory of Clinical and Epidemiological Virology, Rega Institute for Medical Research, Department of Microbiology, Immunology and Transplantation, Catholic University of Leuven, Leuven, Belgium, **12** Université Montpellier, IRIM CNRS UMR 9004 Montpellier, France, **13** Department of Human Genetics, CHU of Liege, University of Liege, Liege, Belgium, **14** Laboratory of Experimental Hematology, Institut Jules Bordet, Université Libre de Bruxelles (ULB), Brussels, Belgium

☙ These authors contributed equally to this work.
* anne.vandenbroeke@bordet.be (AVdB); david_hill@dfci.harvard.edu (DEH); marc_vidal@dfci.harvard.edu (MV); fdequiedt@uliege.be (FD); jean-claude.twizere@uliege.be (J-CT)

**Data Availability Statement:** RNA-sequencing data have been deposited in NCBI's Gene Expression Omnibus (Edgar, 2002) and are accessible through GEO accession number

## Abstract

Viral infections are known to hijack the transcription and translation of the host cell. However, the extent to which viral proteins coordinate these perturbations remains unclear. Here we used a model system, the human T-cell leukemia virus type 1 (HTLV-1), and systematically analyzed the transcriptome and interactome of key effectors oncoviral proteins Tax and HBZ. We showed that Tax and HBZ target distinct but also common transcription factors. Unexpectedly, we also uncovered a large set of interactions with RNA-binding proteins, including the U2 auxiliary factor large subunit (U2AF2), a key cellular regulator of pre-mRNA splicing. We discovered that Tax and HBZ perturb the splicing landscape by altering cassette exons in opposing manners, with Tax inducing exon inclusion while HBZ induces exon exclusion. Among Tax- and HBZ-dependent splicing changes, we identify events that are also altered in Adult T cell leukemia/lymphoma (ATLL) samples from two independent patient cohorts, and in well-known cancer census genes. Our interactome mapping

GSE146210 (https://www.ncbi.nlm.nih.gov/geo/query/acc.cgi?acc=GSE146210).

**Funding:** Computational resources have been provided by the Consortium des Équipements de Calcul Intensif (CÉCI), funded by the Fonds de la Recherche Scientifique (FRS-FNRS, Belgium) under Grant No. 2.5020.11 and by the Walloon Region. This work was primarily supported by the FRS-FNRS grants PDR 14461191 and Televie 30823819 to J-C.T; Fund for Research Training in Industry and Agriculture grants 24343558 and 29315509 to C.V.; Flanders Research Foundation grant # G0D6817N and KU Leuven grant ("Vaast Leysen Leerstoel") to J.V.W; and National Institutes of Health grants P50HG004233 to M.V. and U41HG001715 to M.V., D.E.H., and M.A.C.; M.V. and F.D. are Chercheurs Qualifiés Honoraires, and J.C.T. a Maitre de Recherche of the F.R.S.-FNRS. The funders had no role in study design, data collection and analysis, decision to publish, or preparation of the manuscript.

**Competing interests:** The authors have declared that no competing interests exist.

approach, applicable to other viral oncogenes, has identified spliceosome perturbation as a novel mechanism coordinated by Tax and HBZ to reprogram the transcriptome.

## Author summary

Tax and HBZ are two viral regulatory proteins encoded by the human T-cell leukemia virus type 1 (HTLV-1) via sense and antisense transcripts, respectively. Both proteins are known to drive oncogenic processes that culminate in a T-cell neoplasm, known as Adult T cell leukemia/lymphoma (ATLL). We measured the effects of Tax and HBZ on host gene expression pathway by analyzing the interactome with cellular transcriptional and post-transcriptional regulators, and the transcriptome and mRNA splicing of cell lines expressing either Tax or HBZ. We compared our results with data obtained from independent cohorts of Japanese and Afro-Caribbean patients, and identified common splicing changes that might represent clinically useful biomarkers for ATLL. Finally, we provide evidence that the viral protein Tax can reprogram initial steps of the T-cell transcriptome diversification by hijacking the U2AF complex, a key cellular regulator of pre-mRNA splicing.

## Introduction

The ability of a retrovirus to transform its host cell was originally attributed to integration of retroviral DNA into the host cell's genome. This integration allowed the discovery of cellular oncogenes and related cellular signaling pathways such as the SRC [1], EGFR [2] MYC [3] RAS [4], and PI3K pathways [5]. However, no universal model has been developed to explain oncogenic transformation as a phenotypic result of retroviral integration. The only transforming retrovirus identified in humans to date is the human T-cell leukemia virus type 1 (HTLV-1), which causes adult T-cell leukemia/lymphoma (ATLL). ATLL has a long latency period of approximately 20–60 years, which suggests the occurrence of rare and complex genomic changes during disease progression [6,7]. Proviral integration sites for HTLV-1 are enriched in cancer driver genes, resulting in altered transcription of those genes [8]. Additional genomic changes have also been observed in distant genes that play a role in various mechanisms of T-cell signaling [6,7]. However, the key initial drivers of ATLL are the viral proteins Tax and HBZ, which can independently induce leukemia in transgenic mouse models [9,10].

Studies of Tax and HBZ have been summarized in several reviews [11–15]. These data have been compiled into a KEGG pathway (hsa05166) which highlights the ability of Tax and HBZ to interfere with at least one component of each of the twelve signaling pathways regulating the three cancer core processes: cell fate, cell survival, and genome maintenance [16]. The effects of Tax and HBZ are mediated primarily via protein-protein interactions, and positive or negative transcriptional regulation [14,17]. Tax and HBZ often act in opposing directions in order to control the host's immune response and sustain long-term malignant transformation [18,19].

Our molecular understanding of genomics and transcriptomics deregulation following HTLV-1 infection has come from studying ATLL patient samples [6,20,21]. Because Tax and HBZ show different expression kinetics during ATLL progression [22], it has remained challenging to systematically analyze the relative contribution of each viral protein in reprogramming the host cell's transcriptome and proteome. Here, we carried out a systematic

identification of the interactome networks between Tax/HBZ and cellular regulators of gene expression. We then measured the effects of Tax and HBZ on gene expression at both the transcriptional and post-transcriptional levels. Integration of these interactome and transcriptome datasets provides mechanistic insights into HTLV-1 infection-associated alternative splicing events and leukemogenesis.

## Results

### A comparative interactome of Tax and HBZ with cellular host proteins

Previous studies have shown that Tax and HBZ viral proteins control viral gene expression by competing for binding to key transcriptional factors of the CREB/ATF pathway, and coactivators CBP and p300 (S1A and S1B Fig), complexes that are also specifically targeted by other viral oncoproteins including high-risk human papillomavirus (HPV) E6 proteins [23]. In the present study, we aimed at providing an unbiased map of protein-protein interactions (PPIs) established by Tax and HBZ viral oncoproteins with cellular gene expression regulators, including transcription factors (TFs) and RNA-binding proteins (RBPs) (S1C Fig). Our rationale is that exploring more broadly the similarities and differences of Tax and HBZ interactomes with TFs and RBPs would provide global and specific insights on how viral oncogenes cooperate in the initiation and maintenance of cancer.

Based on Gene Ontology and literature curation, we first established a library that covers 3652 ORFs encoding 2089 transcriptional and 1827 post-transcriptional regulators, including known DNA- and RNA-binding proteins (S1D Fig). Secondly, we assembled a mini-library of different *Tax* (S2A Fig) and *HBZ* (S2B Fig) clones with previously described functional domains (S2A and S2B Fig). We then tested binary interacting pairs between viral products and human TFs and RBPs using our well-established binary interactome mapping strategy employing primary screening by yeast two-hybrid (Y2H) assays, retesting by Y2H and validation using an orthogonal protein complementation assay [24–27]. This interactome search space encompassing ~95,000 binary combinations (S1F Fig), allowed identification of 53 and 116 Tax and HBZ cellular partners, respectively (Fig 1A, S1 Data). Interestingly, we observed a highly significant overlap between gene expression regulators, either TFs or RBPs, interacting with Tax and HBZ (Fig 1A, 25 shared interactors, Fisher test: $p < 2.2e^{-16}$). We next used 90 interacting pairs in a validation experiment using an orthogonal assay, the *Gaussia princeps* protein complementation assay (GPCA) [28]. The validation rate was 83% (Fig 1B and 1C), demonstrating the high quality of our dataset (Y2H_2020) that represents an increase of 32% and 51.6% of known TFs and RBPs interacting with Tax, respectively (Fig 1D). Compared to Tax, fewer interactions were available in the literature for HBZ, and our result represents a substantial increase of 65.5% and 93.6% of TFs and RBPs interacting with HBZ, respectively (Fig 1D).

We expanded our catalog of systematically determined binary Tax and HBZ interactions with host proteins (Y2H_2020) with high quality interactions reported in the literature (Lit_2020) (Figs 1D and S3). The union of Tax-host and HBZ-host interactomes contains 258 and 160 interacting partners, respectively (S3 Fig and S1 Data). Of interest, TFs represent 38% and 59%, while RBPs account for 16% and 38% of Tax and HBZ host interactors, respectively (S3 Fig and S1 Data).

To classify Tax and HBZ interacting partners in functional categories we examined their repartition in the MSigDB hallmark gene sets [29] (Fig 1E). We focused on five categories of specific gene sets (Immune, Proliferation, Signaling, Pathway and Cellular component) and obtained a constructive view of the functional distribution of Tax and HBZ partners. The most significantly enriched gene set signatures were "Proliferation" and "Signaling pathways", for

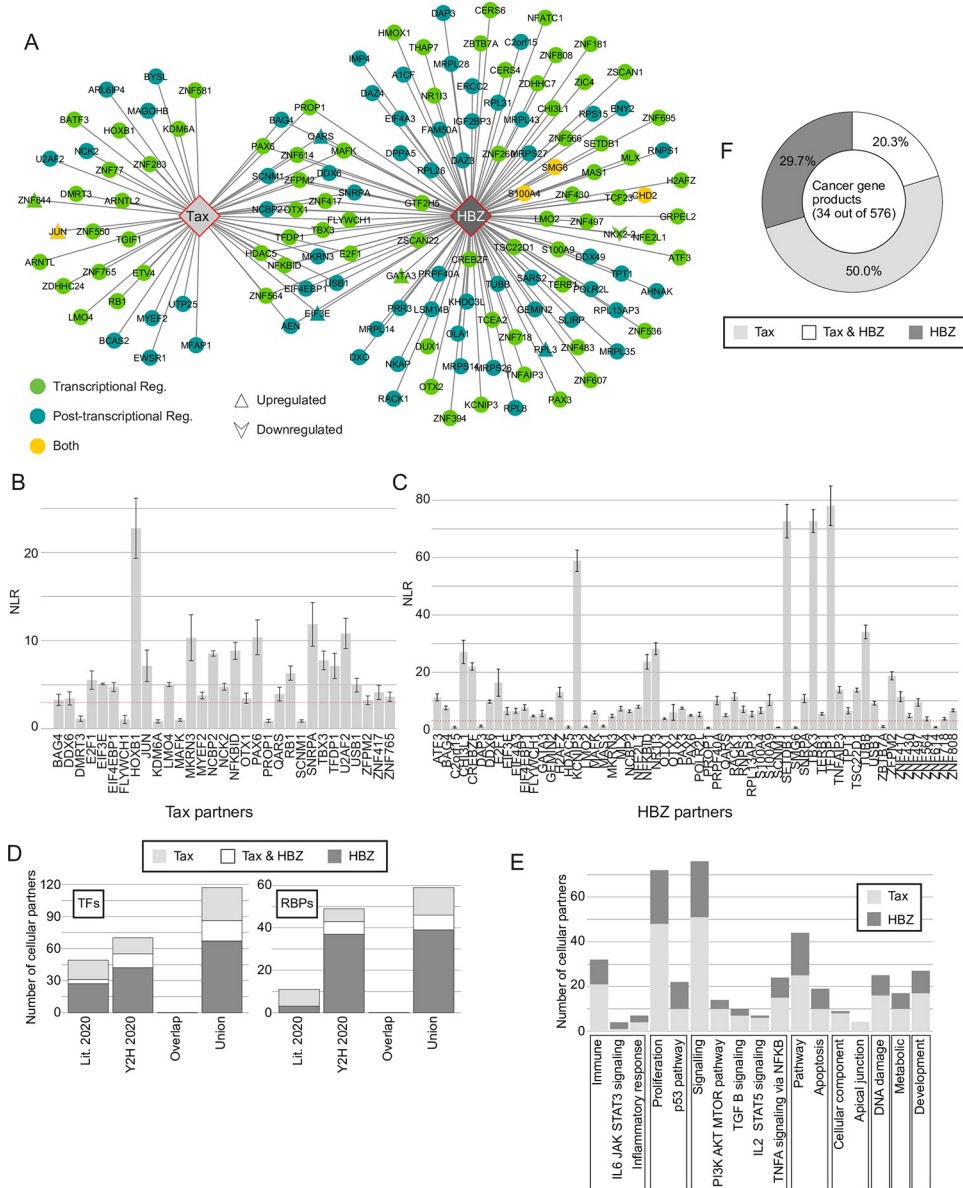

**Fig 1. A comparative interactome network of Tax and HBZ with cellular host genes. (A)** Y2H Interactome map of Tax and HBZ with host TRs (blue), PTRs (green) or both TR/PTR (yellow). Upward triangles and downward arrows show PPIs for which corresponding genes are upregulated or downregulated following Tax or HBZ expression in Jurkat cells. **(B)** and **(C)** GPCA validation assay for Tax (B) or HBZ (C) and their interacting partners identified in Y2H. Y-axis shows normalized luciferase ratios (NLR). Bar graphs represent the mean number ± SD. Positive interactions are indicated by NLRs above dashed red line **(D)** Graph showing the number of RBPs and TFs interacting with Tax and HBZ in different datasets. "Overlap" means interactions already known in the literature, "Union" means the combined list of interactions from this study and the literature **(E)** As in (D) but number of coding genes that are parts of a GSEA hallmark category. **(F)** Percentages of Cancer gene products interacting with Tax and HBZ. See also S1, S2 and S3 Figs and S1 Data.

both Tax and HBZ (Fig 1E). While HBZ appears to specifically target the IL-6-JAK-STAT3 pathway, cancer-related gene products of the PI3K-AKT-mTOR, TGF-β and IL-2-STAT5 pathways are more enriched in the Tax interactome (Fig 1E). Further highlighting the impact of Tax and HBZ in the initiation and maintenance of cancer, these viral proteins exhibit

significantly more PPIs (at least 20 times) than the average "degree" (number of interactors) of known cancer gene products (Fig 1F), or other tumor virus proteins, but comparable to highly connected cancer gene products such as CREB3L1 [24], or HPV viral proteins E6 and E7 [23].

Altogether these results provide strong evidence that Tax and HBZ perturb the cell host through similar and differential associations with transcriptional and post-transcriptional regulators.

## A comparative analysis of transcriptomic changes associated with Tax and HBZ expression

Based on the observation that Tax and HBZ target numerous host DNA- and RNA-binding proteins, it is anticipated that a significant number of host gene expression changes, indispensable for ATLL pathogenesis, are driven by these interactions. We generated a homogeneous inducible cellular system, which consists of two Jurkat T cell lines, Jurkat-iTax and Jurkat-iHBZ, expressing either Tax or HBZ from a doxycycline-inducible promoter (Fig 2A). We then performed high-throughput RNA sequencing (RNA-seq) of the two cell lines and analyzed differentially expressed genes (DEGs), and alternative splicing events (ASEs) associated with Tax or HBZ expression (Fig 2A). We confirmed the induction of Tax and HBZ (Figs 2B, S4A and S4B), which are associated with increased expression of their common up- or down-regulated target genes *GATA3* (Fig 2B) or *SYT4* (S4E Fig), respectively [6,30,31], and differentially regulated *STAT5A* (S4F Fig) [32,33]. Comparative analysis with control cells revealed 1453 and 1014 genes that were differentially expressed upon inducing Tax or HBZ expression, respectively (FDR adjusted p< 0.01 and |log2 fold-change| $\geq$ 1, Fig 2C and 2D and S2 Data). Consistent with its well-known function as a transcriptional activator [34,35], we found that Tax expression caused up-regulation of 885 genes and down-regulation of 568 genes (Fig 2C). In contrast, expression of HBZ was associated with up-regulation of 397 genes and down-regulation of 617 genes (Fig 2D). Several studies found that HBZ can act as a transcription repressor [36–41], in agreement with our finding that 61% of the differentially regulated genes in Jurkat-iHBZ were down-regulated. This finding is also supported by our confocal and transmission electron microscopy observations showing that HBZ-expressing cells have rounder and more regular nuclear speckles (S4G and S4H Fig), which is often associated with decreased (or reduced) Pol II–mediated transcription [42–44].

Gene Set Enrichment Analysis [45] identified TNF-α signaling via NF-kB, as a specific pathway enriched both in Tax- and HBZ-expressing cells (Fig 2E and 2F). In contrast, inflammatory response was specifically up-regulated in Tax-expressing cells whereas cell cycle G2M checkpoint genes were specifically up-regulated in HBZ expressing cells (Fig 2E and 2F and S2 Data). Interestingly, the extent of co-regulated genes by both viral proteins was highly significant (28% for Tax and 41% for HBZ, empirical P < 0.0001). These include 405 genes whose expression was altered in the same direction (149 up-regulated and 256 down-regulated) and only 12 genes whose expression was altered in opposite directions (S2 Data). Despite their differential expression *in vivo*, our transcriptomic changes analysis confirms the notion highlighted above, that Tax and HBZ viral proteins drive a number of overlapping molecular perturbations.

## Impact of Tax and HBZ expression on cellular gene alternative splicing events

Analyses of Tax and HBZ host cell perturbations have predominantly focused on transcriptional effects. However, from our interactome data, the interacting proteins annotated as "post-transcriptional regulators" represent 54% and 44% of Tax and HBZ partners involved in

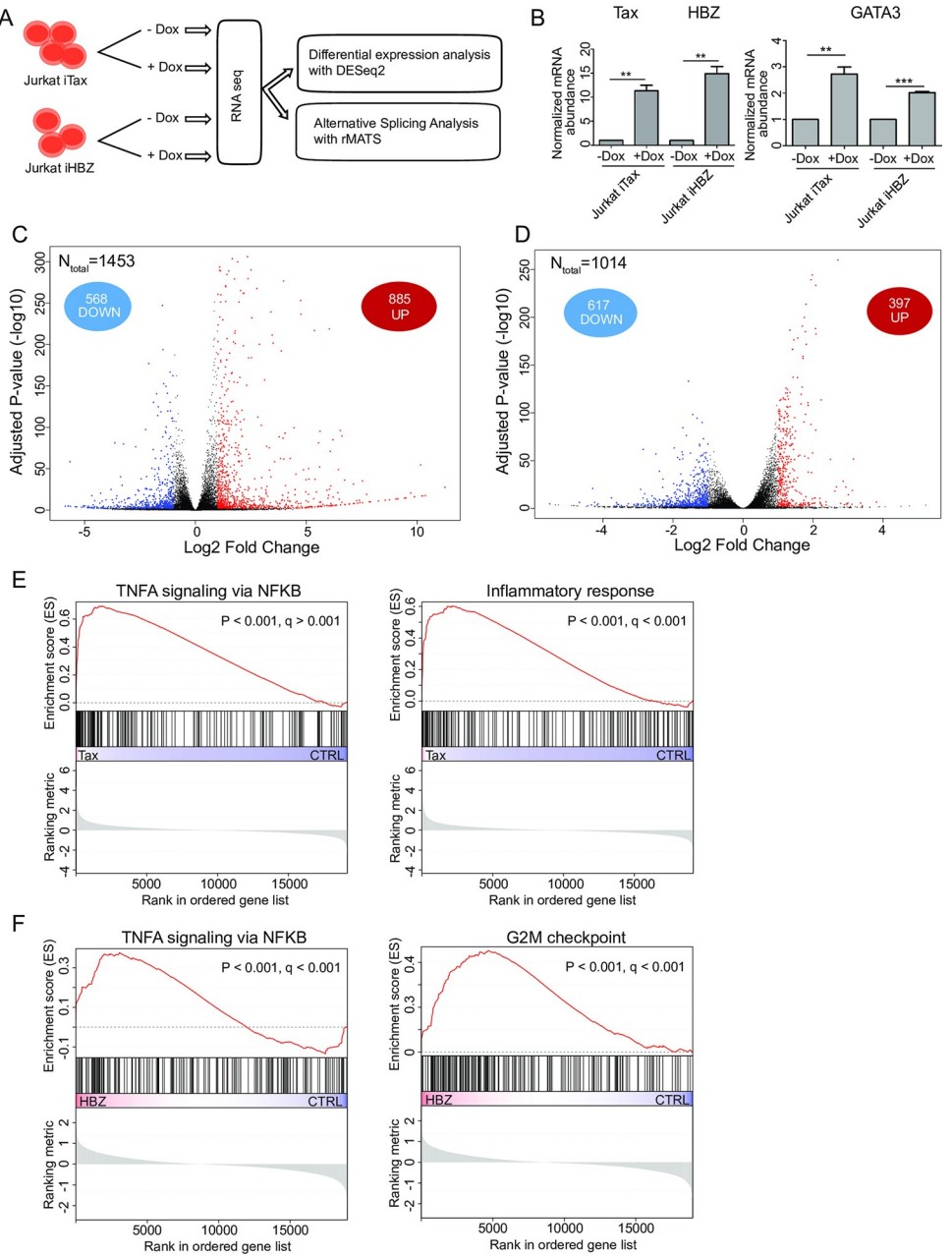

**Fig 2. A comparative analysis of transcriptomic changes associated with Tax and HBZ expression. (A)** Overview of the experimental design for analysis of abundance and splicing alterations upon Tax and HBZ expression in Jurkat cells. Three replicates of each condition were used for RNA seq analysis. **(B)** Tax and HBZ expression was analyzed by qRT-PCR in Jurkat cells after induction with doxycycline (left). Overexpression of GATA3 in Jurkat-iTax and Jurkat-iHBZ cell lines (right). ** = p-value <0.01, *** = p-value <0.001 **(C-D)** Volcano plot of significantly up-regulated genes (red dots) and down-regulated genes (blue dots) upon induction of Tax (C) or Flag-HBZ (D) in Jurkat cells. **(E-F)** Examples of significantly enriched gene sets by GSEA in Jurkat-iTax (E) cells and Jurkat-iHBZ (F) cells. See also S2 Data.

gene expression regulation, respectively (S3 Fig). To assess the impact of Tax and HBZ on the cellular splicing landscape, we statistically computed splicing events using the rMATS software (Fig 3A, [46]). We focused on 5 types of alternative splicing: Skipped Exons (SE) or cassette

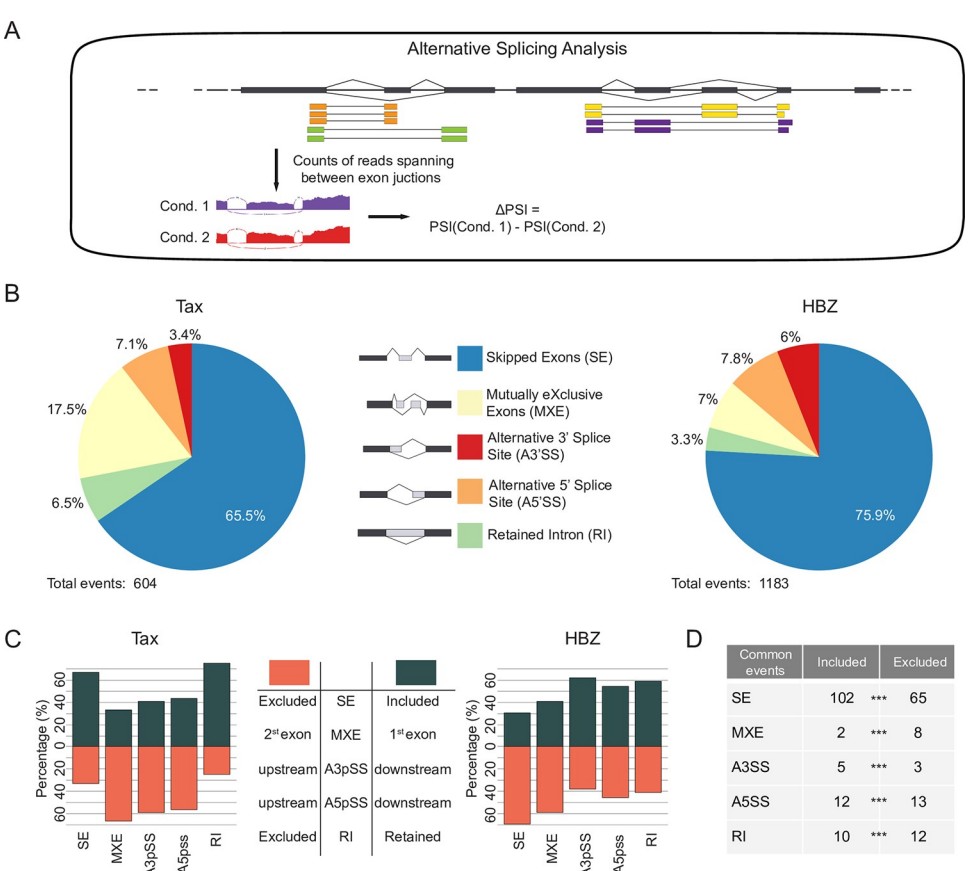

**Fig 3. Impact of Tax and HBZ expression on cellular gene alternative splicing events. (A)** Differential alternative splicing between Tax or HBZ expressing conditions and control conditions were analysed with rMATS (n = 3 for each condition). Following alignment with STAR, exon inclusion levels are assessed using reads that are mapped to both regulated exons/introns and spliced junctions. rMATS determines the Percent Spliced In (PSI) in each condition and evaluates the difference in PSI (ΔPSI) of alternative splicing events. Significant events were then user-filtered first by discarding genes with low TPM and keeping AS events with a |ΔPSI| ≥0.1 (i.e. 10%) and a FDR <0.05. **(B)** Splicing profiles of ASEs detected in Jurkat cells expressing Tax (left) or Flag-HBZ (right). SE = Skipped Exon, MXE = Mutually Exclusive Exons, A3'SS = Alternative 3' Splice Site, A5'SS = Alternative 5' Splice Site, RI = Retained Intron. **(C)** Exclusion or inclusion of ASEs observed in Tax and Flag-HBZ expressing conditions. **(D)** Number of shared ASEs between Jurkat cells expressing Tax or Flag-HBZ. *** = p-value <0.001. See also S4 Fig and S3 Data.

exons, Mutually Exclusive Exons (MXE), alternative 3' Splice Sites (3'SS), alternative 5' Splice Sites (5'SS) and Retained Introns (RI) [46], which were quantified as Percent Spliced In (PSI). Significantly different AS events were selected by filtering events with a ΔPSI ≥0.1 or ≤-0.1 and a false discovery rate (FDR) <0.05. (Fig 3A and S3 Data). Our analysis identified 604 Tax- and 1183 HBZ-regulated alternative splicing events (ASEs) corresponding to 447 and 839 genes, respectively (Fig 3B). The majority of Tax and HBZ-dependent events were cassette exons (65.5% and 75.9% for Tax and HBZ, respectively) (Fig 3B), as previously reported [47].

Interestingly, Tax and HBZ had globally opposite effects on the cellular splicing landscape (Fig 3C). This was particularly true for SE where most regulated exons (66.9%) showed increased inclusion upon Tax expression. In contrast, the majority of HBZ-regulated cassette exons (69.3%) were more likely to be skipped. However, we also found a significant overlap between the different Tax- and HBZ-regulated ASEs: 167 SE (P = 3.51e-130), 8 A3'SS (P = 4.17e-12), 25 A5'SS (P = 3.24e-31), 10 MXE (P = 2.10e-09) and 22 RI (P = 7.15e-20) (Fig 3D). As observed for differentially expressed genes, a number of SE exons were similarly

regulated by Tax and HBZ expression (102 inclusion and 65 exclusion events in both Tax and HBZ expressing cells, Fig 3D). Taken together our analysis shows that although Tax and HBZ have globally opposite effects on the host splicing landscape, they share some common target exons that are similarly affected.

## Splicing targets of Tax and HBZ are enriched for cancer-related genes

We validated the rMATS analysis of 10 Tax- and 13 HBZ-dependent splicing events by qRT-PCR (10/10 for Tax, 12/13 for HBZ, Fig 4A), and performed Gene Ontology (GO) enrichment analysis. Splicing targets were significantly enriched by several GO terms in HBZ-regulated genes, whereas no significant enrichment was detected for Tax splicing targets. For HBZ, enriched categories were mostly related to RNA regulation, especially RNA splicing (S3 Data). We further investigated the functions of specific spliced genes in cells expressing Tax or HBZ by examining their repartition in the MSigDB hallmark gene sets built by GSEA [29] (Fig 4B). A significant overlap was also found between genes corresponding to Tax or HBZ splicing pre-mRNA targets and known cancer census genes (ncg.kcl.ac.uk and cancer.sanger.ac.uk) (P = 0.02247 for Tax and P = 0.001742 for HBZ). Pre-mRNA of 33 and 63 cancer genes were identified as splicing targets of Tax and HBZ, respectively (Fig 4C, S3 Data). Among these, 10 were similarly deregulated by Tax and HBZ (*CHCHD7, EIF4A2, NF2, POLG, PTPRC, UBR5, ABI1, BCLAF1, FLNA* and *NSD1*). These include *PTPRC*, coding for CD45, a transmembrane protein tyrosine phosphatase, which is known to be alternatively spliced upon T-cell differentiation and induces a switch from naive (CD45RA) to memory T-cells [48–51].

Another interesting observation here is the fact that only a small number of genes (23 and 29 for Tax or HBZ, respectively) presenting ASEs were differentially expressed (Fig 4D and 4E, S3 Data), as previously observed [47]. Since co-transcriptional processing is a widespread mechanism for many genes in different organisms [52–54], our result suggests that splicing events and transcription are not functionally coupled for the majority of genes regulated by Tax or HBZ.

## ATLL-specific splicing events validated in independent cohorts partially overlap with Tax- and HBZ-driven splicing

To determine whether ASEs detected in Tax or HBZ expressing cells could be relevant for HTLV-1 infection and leukemogenesis, we interrogated RNA-seq data obtained from two independent cohorts. In the first cohort, referred to as "the Japanese cohort", we analyzed 35 ATLL samples, 3 samples from HTLV-1 asymptomatic carriers and 3 samples from healthy volunteers [6]. Using rMATS v3.2.1 [46], we detected 4497 ASEs (in 2343 genes) between HTLV-1 carriers and healthy volunteers, while 9715 events (in 2737 genes) were differentially affected in the ATLL patients compared to healthy volunteers (Fig 5A). Among these, cassette exons and mutually exclusive exons accounted for a large majority of ASEs (Fig 5A), as observed for cells expressing Tax or HBZ proteins (Fig 3B). Interestingly, we observed a significant overlap between ASEs detected in HTLV-1 carriers and ATLL patients (1637 ASEs on 1238 genes, SE P ~0, A3'SS P = 8.026e-112, A5'SS P = 4.14e-118, MXE P = 1.40e-181, RI P = 5.01e-19), suggesting that some ASEs may initiate disease progression and persist during ATLL. Similar to our observations in Jurkat cells expressing Tax, alternative splicing of cassette exons (SE events) was skewed towards inclusion for both HTLV-1 carriers and patients with ATLL (Fig 5B). The most significant GO-term enrichment for genes presenting ASEs in ATLL patients was RNA binding (GO:0003723). However, other categories were also significantly enriched, such as cadherin binding, viral process and immune system process (S4 Data). Thirty-one and 42 ASEs observed in cells expressing Tax or HBZ, respectively, were also

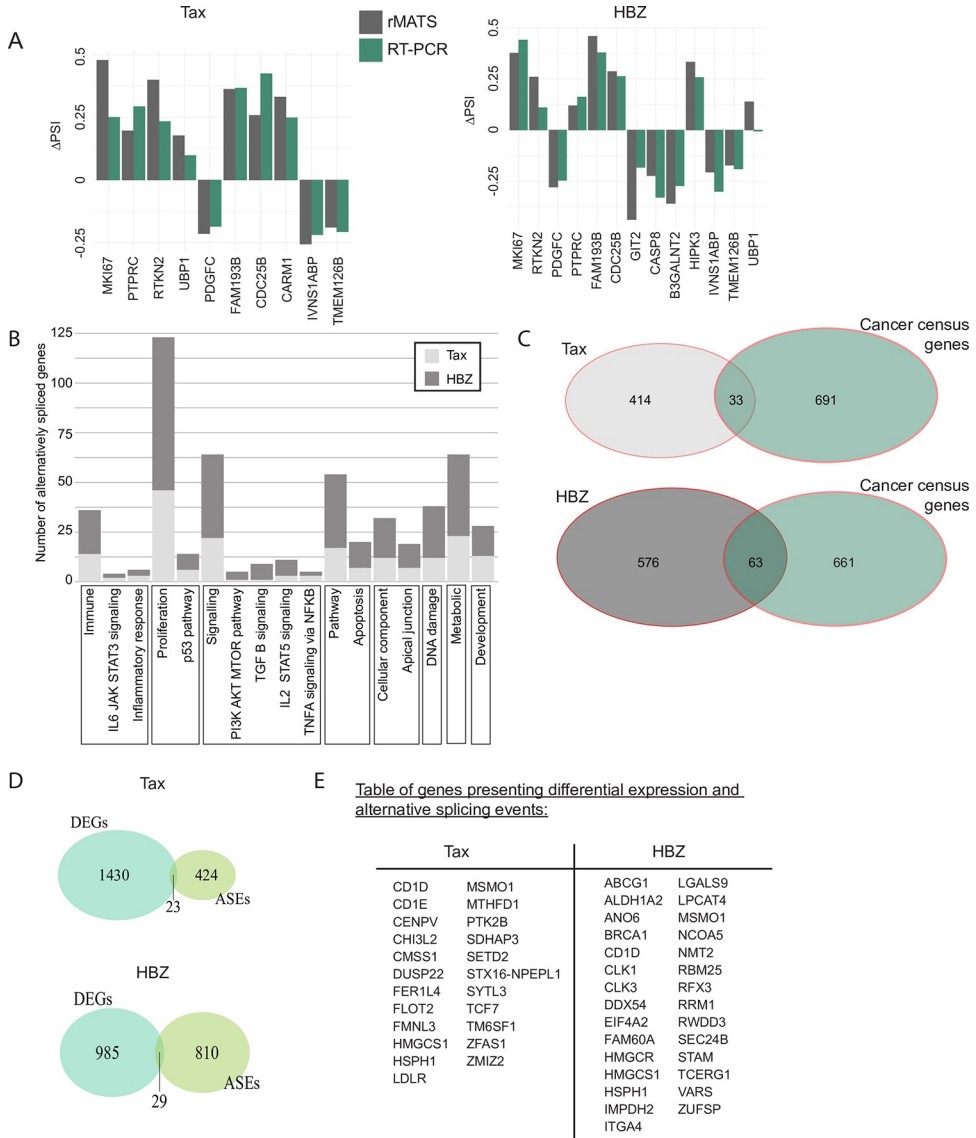

**Fig 4. Splicing targets of Tax and HBZ are enriched for cancer related genes. (A)** Validation of ASE events, 10 for Tax and 13 for HBZ, by RT-PCR. Graphs showing the ΔPSI experimentally determined by RT-PCR and the ΔPSI computationally obtained by rMATS for these events for Tax (left) and HBZ (right) **(B)** Alternatively spliced (AS) genes in iTax or iHBZ cells, which are part of a GSEA hallmark category. Cancer genes are in red. **(C)** Venn diagrams showing coverage of genes presenting ASEs in Jurkat iTax (top) or Jurkat iHBZ (bottom) cells with known cancer census genes. **(D)** Venn diagrams showing coverage between up- or down-regulated genes upon Tax or HBZ expression (blue) and genes with ASEs (green). **(E)** Table of genes with DE and AS in Tax or HBZ expressing cells See also S5 Fig and S3 Data.

present in patients with ATLL. Those events occurred on pre-mRNA of 56 genes including well-known cancer-related genes *PTPRC, IKBKB* and *HRAS*, and genes coding for transcription factors ILF2, ATF2 and EYA3 (Fig 5C and 5D).

In the second cohort, referred to as the "Afro-Caribbean cohort", we analyzed 29 ATLLs, 5 HTLV-1 asymptomatic carriers and 4 CD4+ T-cell samples from uninfected healthy individuals [8,55,56]. Using rMATS v4.1.1 [46], we identified 2826 ASEs in HTLV-1 carriers and 4172 ASEs in ATLL patients (Fig 6A). Similar to the Japanese cohort, the majority of ASEs identified

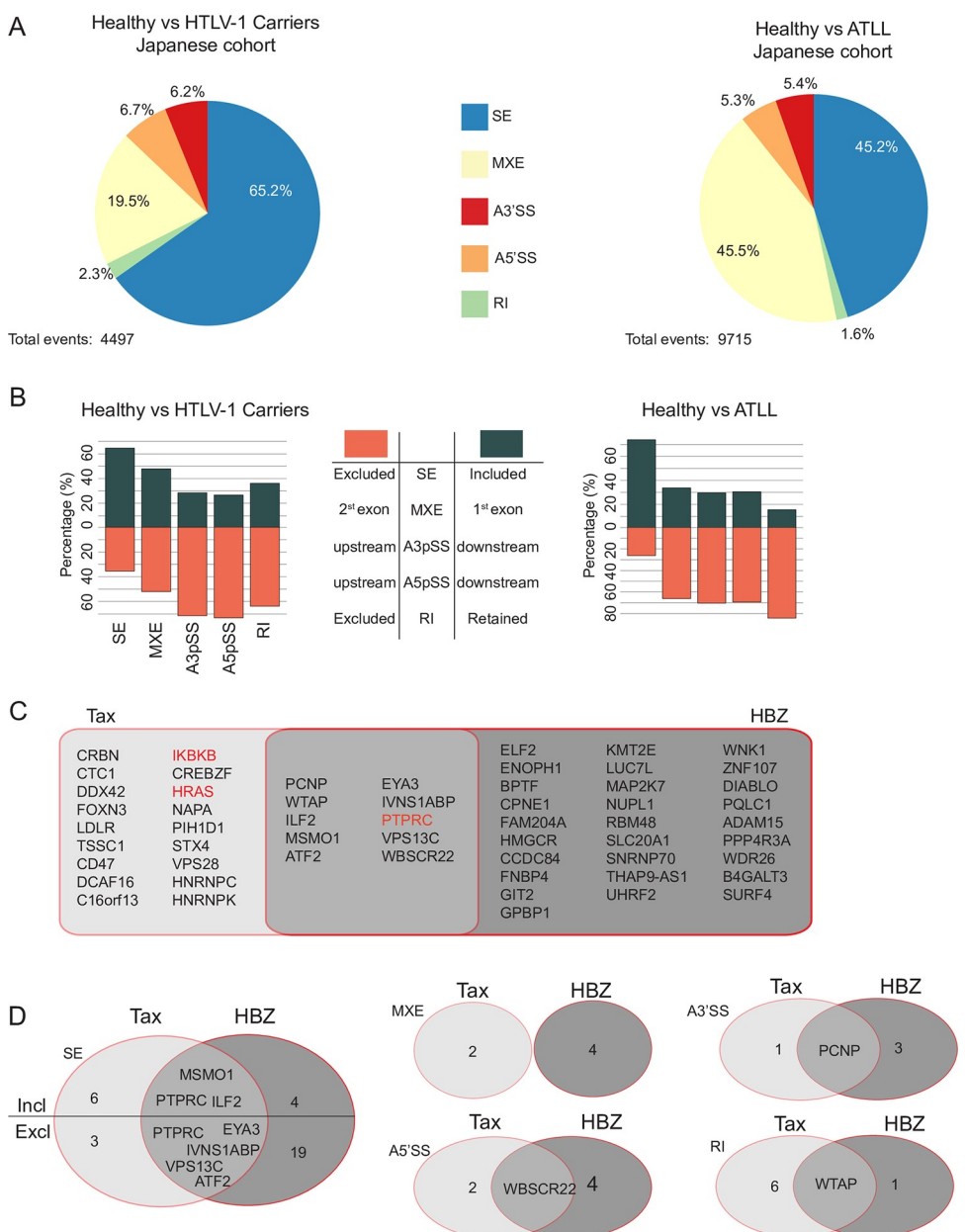

**Fig 5. Splicing events in ATLL patients from the Japanese cohort.** (A) ASEs detected in HTLV-1 carriers or patients with ATLL compared to healthy human peripheral CD4+ T cells. SE = Skipped Exon, MXE = Mutually Exclusive Exons, A3'SS = Alternative 3' Splice Site, A5'SS = Alternative 5' Splice Site, RI = Retained Intron (B) Exclusion or inclusion of ASEs observed in HTLV-1 carriers or ATLL. (C) Genes sharing similar ASEs in ATLL and Jurkat cells expressing Tax or Flag-HBZ. (D) Number of shared events. See also S3 Data.

in Afro-Caribbean patient samples were SE and MXE (Fig 6A), with a higher number of inclusion than exclusion events (Fig 6B). We observed an overlap between ASEs detected in HTLV-1 carriers and ATLL patients (1492 ASEs on 1074 genes. SE P ~0, A3'SS P = 5.62e-114, A5'SS P = 9.48e-64, MXE P = 1.56e-242 RI P = 4.29e-132).

We found that a highly significant number of same genes were affected by ASE in both the Afro-Caribbean and the Japanese cohort (Fig 7A, S3 Data), while some differences were also observed (compare Figs 5C, 5D, 6C and 6D). We identified genes for which ASEs were

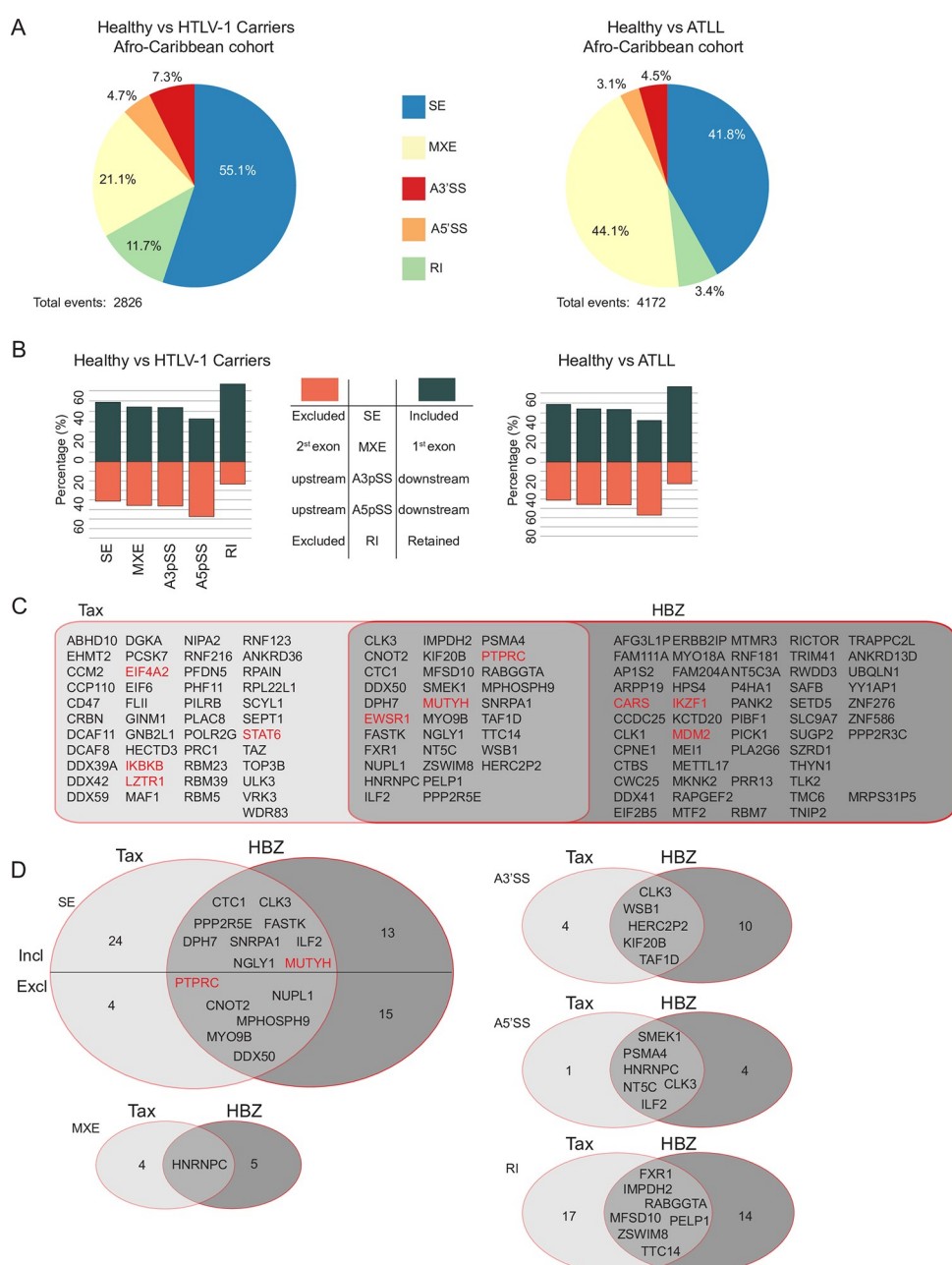

**Fig 6. Splicing events in ATLL patients from the Afro-Caribbean cohort.** (A) ASEs detected in HTLV-1 carriers or ATLL patients compared to healthy human peripheral CD4+ T cells. SE = Skipped Exon, MXE = Mutually Exclusive Exons, A3'SS = Alternative 3' Splice Site, A5'SS = Alternative 5' Splice Site, RI = Retained Intron (B) Exclusion or inclusion of ASEs observed in HTLV-1 carriers or ATLL. (C) Genes sharing similar ASEs in ATLL and Jurkat cells expressing Tax or Flag-HBZ. (D) Number of shared events. See also S3 Data.

observed in both the Afro-Caribbean cohort and the Jurkat cells expressing Tax or HBZ, including cancer genes *EIF4A2*, *IKBKB*, LZTR1, *STAT6* (for Tax); *CARS*, *MDM2*, *IKZF1* (for HBZ); and *EWSR1*, *MUTYH*, and *PTPRC* (for both Tax and HBZ) (Fig 6C and 6D). Several splicing events regulating the inclusion/exclusion of one or more of these exons were detected in both cohorts, thus representing validated molecular biomarkers of ATLL. In addition, we

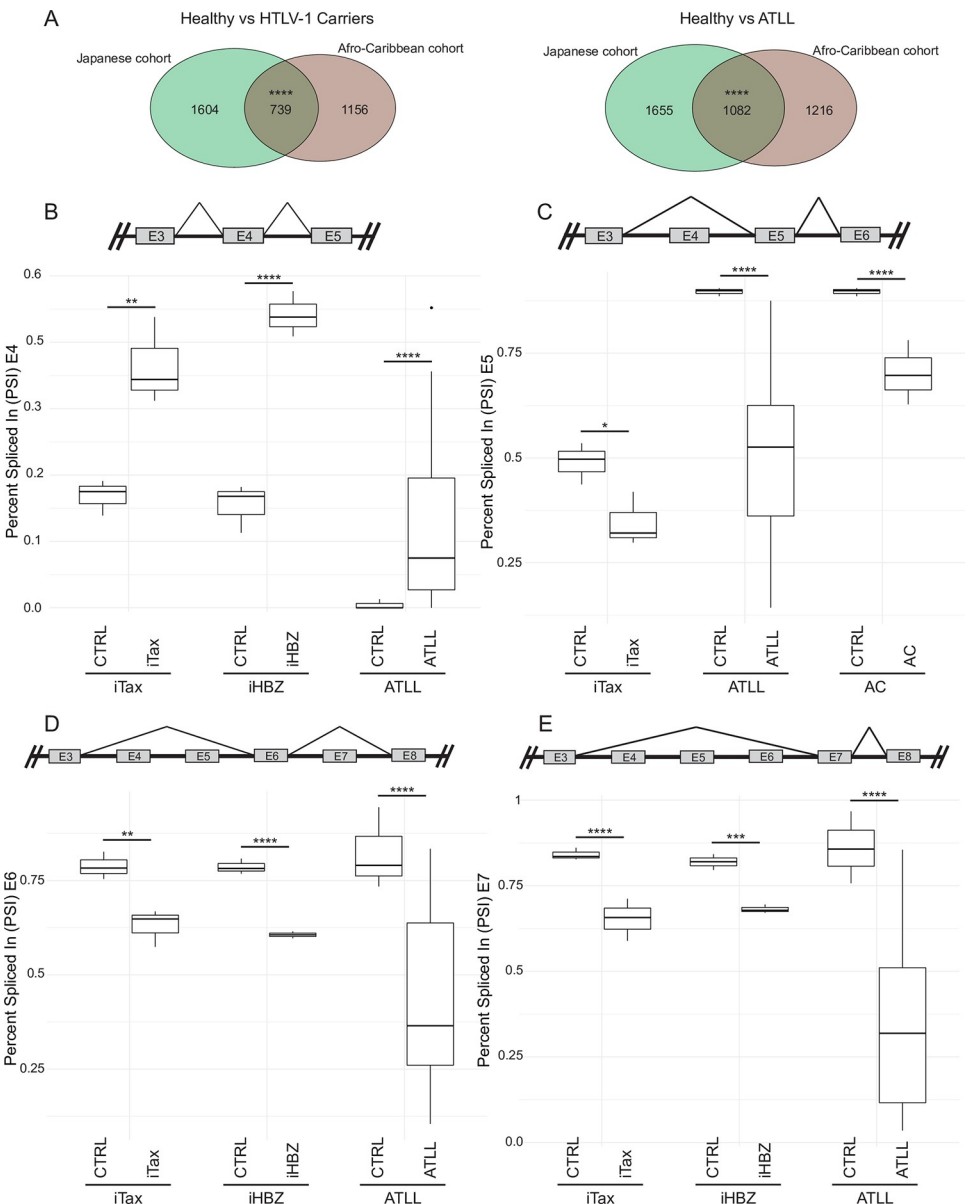

**Fig 7. Alternative splicing events affecting PTPRC exons 4 to 7. (A)** Venn diagrams showing genes affected by ASEs shared between the two patient cohorts, in carriers (left) and ATLLs (right). Significant overlap between HTLV-1 carriers (p = 2.57e-105) and ATLL patients (p = 1.97e-171) of both cohorts calculated by hypergeometric test. **(B-E)** Box plots of PSI values of AS exons from significant SEs detected on exons 4 to 7 of PTPRC in the Japanese cohort. A diagram representing the regulated splicing event is shown above each graph. (B) Increased inclusion of exon 4. Decreased inclusion of exon 5 (C), 6 (D) and 7 (E). PSI values for each replicate and event were computationally calculated by rMATS (n = 3 except for ATLL patients where n = 35). FDR of p-values calculated with rMATS by likelihood ratio test. p-value thresholds depicted as follows: *p<0.05, **p<0.01, ***p<0.001, ****p<0.0001. See also S6 Fig and S3 Data.

observed similar ASEs for each of the affected exons in Tax/HBZ Jurkat cells and patient samples. Notably, several ASEs in the *PTPRC* pre-mRNA coding for pan-leukocyte marker CD45 detected in Tax- and HBZ-expressing cells were also identified in patients with ATLL. These ASE changes affected *PTPRC* exons 4, 5, 6 and 7. More specifically, we observed a trend for exons 5, 6 and 7 to be excluded, while exon 4 was more included (Figs 7B–7E and S6A–S6D),

suggesting that, compared to classical activated and memory T-cells [50], ATLL cells exhibit a specific alternative splicing pattern on *PTPRC* pre-mRNA. Those exons are important for the production of diverse CD45 isoforms and therefore, the inclusion of exon 4 may represent a major mechanism used by Tax and HBZ to control T cell activation. Other examples include *ATF2* and *EYA3*, which present premature stop codons on spliced exons; and *MSM01*, which has a Nuclear Localization Signal (NLS) affected by splicing events (S5 Fig). Altogether, our results demonstrate a global impact of Tax and HBZ expression on alternative splicing. Strikingly, Tax and HBZ targets show different splicing patterns in HTLV-1-infected individuals and ATLL patients, further emphasizing their global opposing effects on the deregulation of host genes.

## Identification of RNA splicing-specific roles for Tax and HBZ proteins

Tax and HBZ interact respectively with 35 and 47 proteins involved in RNA catabolic processes (Fig 8A, yellow), RNA export (Fig 8A, light red), RNA processing (Fig 8A, blue) and/or RNA translation (Fig 8A, green), respectively. RNA processing factors interacting with Tax and HBZ are categorized into pre-mRNA processing and splicing factors (Fig 8B). To further explore the physiological roles for Tax- and HBZ-dependent effects on host mRNA splicing, we first performed motif enrichment analysis of alternative splicing events detected in Japanese patients with ATLL. Regulated exons and part of their flanking introns were screened for RNA-binding motifs using the MEME suite [57]. We found significant enrichment for 31 RNA-binding motifs including the complementary factor for APOBEC-1 (A1CF), an RNA-binding protein regulating metabolic enzymes via alternative splicing [58], identified here as a HBZ partner, and the U2 small nuclear ribonucleoprotein particle (snRNP) auxiliary factor (U2AF) large subunit U2AF65 (also called U2AF2), identified here as a Tax partner (Figs 8C and S7A and S1 Data).

The U2AF complex is a well-established essential component of the spliceosome assembly pathway [59–61]. U2AF65 forms a heterodimer with U2AF35 that recognizes the 3' splice sites (3'SS) of introns [60]. U2AF65 binds to the polypyrimidine tract (PY tract) of the intron and induces the recruitment of the U2 snRNP [60,61]. We performed co-immunoprecipitation assays and confirmed an interaction between Tax and endogenous U2AF65 (Fig 8D). This interaction was dependent on the presence of RNA (Fig 8E). Using the GPCA assay [28], we confirmed direct Tax and U2AF65 interaction, and the absence of binding between Tax and U2AF35 (Fig 8F). However, Tax expression increased the formation of the U2AF35-U2AF65 heterodimer (Fig 8G) suggesting that the interaction interfaces of Tax/U2AF65 and U2AF35/U2AF65 may be different, and Tax does not disrupt, but rather stabilizes, the U2AF complex. As a control for specificity, we did not detect any interaction between the U2AF subunits and HBZ (S7B Fig).

Previous studies have shown that a more stable U2AF35/U2AF65 complex could favor the recognition of less conserved sub-optimal PY tract sequences containing a lower proportion of thymidine (T) [59,62]. To determine if there is a global effect of the Tax/U2AF65/U2AF35 interaction on alternative splicing events, we inspected the 30bp upstream of all 3'SS of cassette exon events, whether altered or not upon Tax expression. We found a significant reduction of T content in the vicinity of 3'SS of more included exons, indicating that exons with less conserved PY tracts are more included following Tax expression (Fig 8H). To further evaluate the interplay between Tax and U2AF65, we generated by shRNA a Jurkat T-cell line with reduced U2AF65 expression (Fig 8I). We analyzed 6 SE events detected in Jurkat-iTax cells, and as shown on Fig 8J-O, knockdown of U2AF65 affected splicing of all exons, and induced higher inclusion levels for 5 out 6 SE events.

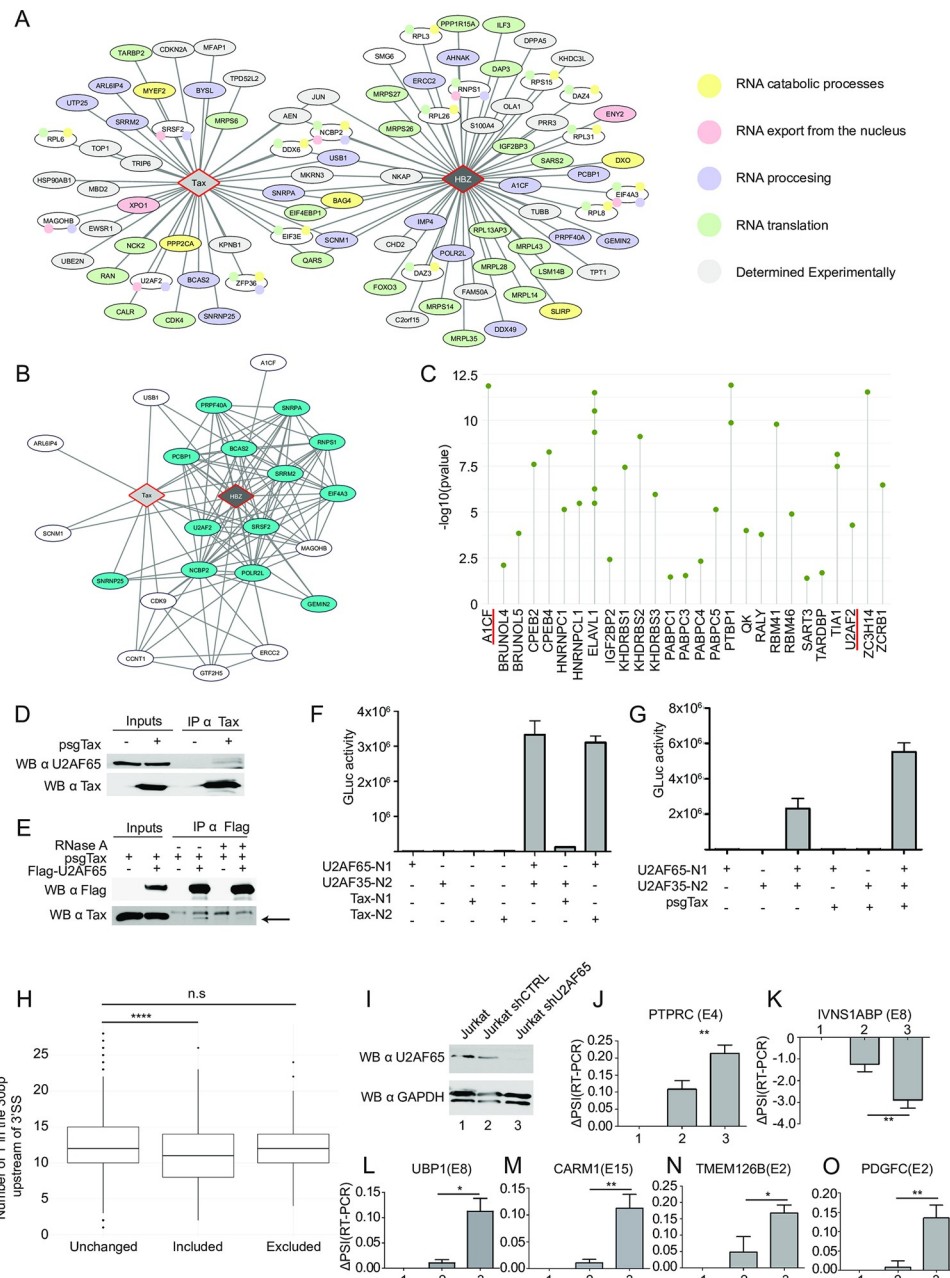

**Fig 8. Identification of RNA splicing-specific roles for Tax and HBZ proteins.** (**A**) Interactome of Tax and HBZ with PTRs, color-coded according to their GO categories. (**B**) Proteins involved in splicing (blue) are represented by wide edges. Thinner edges represent interactions between host proteins. (**C**) RNA-binding protein motif enrichment analysis of ASEs detected in ATLL samples. Sequences of regulated exons and part of their flanking introns were screened for RNA-binding motifs using AME from the MEME suite. Enriched motif(s) of RBPs with an adjusted p-value <0.05 are shown. Known partners of Tax or HBZ are highlighted in red. (**D**) Co-Immunoprecipitation of Tax and U2AF65 in HEK293T cells. (**E**) Co-Immunoprecipitations of U2AF65 and Tax, in HEK293T cells treated or not with RNAse A. Detection of Tax is shown by an arrow. (**F**) GPCA assay for Tax and U2AF complex subunits. Y-axis shows average luciferase activities of three independent experiments. (**G**) GPCA assay, in HEK293T cells, for U2AF complex subunits under expression of Tax. Y-axis shows average luciferase activities of three independent experiments. (**H**) Number of T in the 30bp upstream of the 3'SS of alternatively spliced exons in Tax expressing cells. Medians are represented by a line. P = 4.044e-08. (**I**) Immunoblotting of U2AF65 in Jurkat cells knocked down for *U2AF65* expression and control cells. (**J-O**) ΔPSI of AS exons in both conditions. Regulated exons are indicated in brackets. * = p-value <0.05, ** = p-value <0.01. See also S7 Fig.

In conclusion, although other RNA processing factors interacting with Tax (Fig 8A and 8B) are likely to influence the U2AF heterodimer and therefore splice site recognition, our results demonstrate that a large part of Tax-driven SE events are U2AF65-dependent, suggesting that, the stabilization of the U2AF complex by Tax potentially drives the initial steps of transcriptome diversification in HTLV-1 leukemogenesis.

## Discussion

Splicing events, producing multiple mRNA and protein isoforms, participate in proteome diversity and contribute to phenotypic differences among cells [63]. Splicing programs are often altered in cancer cells and systematic quantification of splicing events in tumors has led to the identification of cancer-specific transcripts that are translated into divergent protein isoforms participating in oncogenic processes [63]. In the context of infectious diseases, it is well known that viruses exploit the host splicing machinery to compensate for their small genomes and expand the viral proteome [64]. As a consequence, interactions with host RNA splicing factors have been reported for a number of viruses, including the Human Immunodeficiency [65], Influenza A [66], Herpes Simplex [67], Epstein-Barr Virus [68], Reovirus [69], Human Papillomaviruses [70,71], human Adenovirus [72], or human Parvovirus [70]. However, there is very limited understanding of the direct or indirect effects of viral products in regulating host RNA splicing.

Tax and HBZ, two HTLV1-encoded proteins that are major drivers of ATLL, have been known for many years to hijack the host gene expression programs. However, to date, they are exclusively considered as transcriptional regulators, acting at the level of mRNA synthesis. In HTLV-1 positive cells, the *HBZ* antisense transcript is consistently expressed while Tax-dependent sense transcription occurs in a burst-like manner, allowing HTLV-1 transmission to naïve T- cells [22]. We thus performed high-throughput binary interactome mapping to identify novel interacting partners of Tax and HBZ and generated a homogenous expression model based on independent induction of Tax and HBZ expression in a Jurkat T-cell line. Jurkat is an established T-cell acute lymphoblastic leukemia cell line [73], which, like any in vitro culture system, carries genomic and phenotypic differences when compared to primary T cells isolated from healthy donors or HTLV-1 infected peripheral blood mononuclear cells. Although our Jurkat experimental system may be limited in capturing the *in vivo* physiological effects of Tax and HBZ in primary cells, it does allow for a systematic comparative analysis for identifying Tax- and HBZ-dependent events contributing to the dysregulation of cellular functions.

First, we systematically identified, for Tax and HBZ viral proteins, shared and distinct human interacting partners implicated in gene expression regulation. Although we have not interrogated post-translational modification-dependent interactions, this map is the first to be reported for HTLV-1 and constitutes a valuable resource for in depth analysis of Tax and HBZ molecular functions. Our data suggest that both viral proteins interfere almost equally with all steps of mRNA life, including splicing processes, to reprogram the host cell transcriptome. Second, we used a Jurkat T-cell line model to identify potential Tax and HBZ splicing targets, which were validated in primary cells isolated from two independent cohorts of HTLV-1 asymptomatic carriers and ATLL patients [6,8,55,56]. These two cohorts are unbalanced in the number of control samples (purified CD4+ T-cells) from healthy individuals compared to HTLV-1 positive samples, and their genetic and epidemiological heterogeneity could affect our results. However, analyzing these two cohorts separately allowed us to draw three conclusions: (i) in both cohorts, spliced exons (SE) and mutually exclusive exons (MXE) accounted for a large majority of ASEs (compare Figs 5A and 6A), (ii) a number of ASEs from both

cohorts were also identified in Jurkat cells expressing either *Tax* or *HBZ* genes (compare Figs 5C and 6C) and (iii) we have identified common ASEs in both patient cohorts that partially overlap with *Tax*- or *HBZ*-driven splicing events (Fig 7A). Interestingly, Tax- and HBZ-dependent splicing events also affected 33 and 63 genes that are also included in the Catalogue of Somatic Mutations in Cancer (COSMIC), as part of the cancer gene census [74]. One particular interesting example is the *PTPRC* gene encoding CD45, a critical regulator of immune cell development [48]. The relative inclusion of exons 4 to 7 of *PTPRC* was modified in Tax and HBZ expressing cells, as well as in ATLL patients' samples (Fig 7). These alternative splicing events on the pre-mRNA of the *PTPRC* gene have been shown to lead to the expression of different CD45 isoforms [75–77] dictating immune cell development [48]. However, the mechanisms regulating CD45 isoform expression are not well understood. Protein kinase C (PKC) has been shown to induce PTPRC exon exclusion in a T-cell model [78], which correlates with previously shown activating mutations in PKC genes [6] and exclusion of exons 5, 6 and 7 in ATLL samples reported here (Fig 7). In order to propose an ATL-specific CD45 isoform as a potential diagnostic tool, future studies will be needed to address (i) mRNA and protein isoform expression at the single-cell level during disease progression, (ii) identification of specific extracellular ligands and possibly (iii) segregation of the functional interplay between Tax and HBZ in regulating PTPRC splicing.

Other examples from this study include well-described tumor-promoting genes such as *SRSF2*, *DNMT3A*, *ATM*, *BRCA1* and *PKM* [63], for which Tax- and HBZ-alternative splicing events are now associated with ATLL for the first time. Information about specific splicing isoforms of these genes would be beneficial to our understanding of ATLL biology, including perturbation of metabolic pathways. For instance, *PKM* pre-mRNA exhibits MXE changes in HTLV carriers (S3 Data). The encoded enzyme, pyruvate kinase PKM, is essential in glycolytic ATP production [79]. Furthermore, we found that pre-mRNAs whose splicing is modified following HBZ expression are significantly enriched in metabolic processes (S3 Data), and showed that HBZ interacts with A1CF, an RNA-binding protein that regulates metabolic enzymes via alternative splicing [58], and for which we observed binding motif enrichment in ATLL samples (Fig 8C). We propose that HBZ, via interactions with RNA-processing factors, controls metabolic pathways leading to the maintenance of infected cells in a low glucose concentration and highly hypoxic host microenvironment, as recently described [80]. It will be interesting to explore whether inhibitors/activators of HBZ interactions with specific splicing factors targeting metabolic pathways may reveal novel therapies for ATLL. Lastly, this study identified genes such as *IKZF1-5*, *CLK1*, *FRS2*, *HNRNPD*, *HSPH1*, *ITGAE*, *MKI67*, *RTKN2* and *SAT1* that could be used as potential biomarkers for ATLL because they are shared between our study using, Jurkat cells, and a dataset from untransformed infected CD4+ T-cell clones and ATLL samples [20,81].

## Concluding remarks

Just as the identification of major transcription factors interacting with Tax and HBZ (CREB/ATF, AP-1, β-catenin, Smad and NF-kB) revolutionized our understanding of the mechanisms underlying HTLV-1 pathogenesis, our finding that both viral proteins are also able to manipulate post-transcriptional events provides an excellent opportunity for an in-depth analysis of less-explored gene expression events (RNA catabolism, export, splicing and translation).

Of broad interest, our work is a contribution toward the cancer genome atlas exploration that has already revealed alternative splicing events across 32 cancer types and highlighted the U2AF complex as a key player in cancer transcriptome diversification [82,83]. As shown here, the retroviral proteins Tax and HBZ are excellent tools to analyze transcriptional dynamics in cancer cells, beyond the evaluation of mutated genes. We advance the hypothesis that the viral

protein Tax could reprogram initial steps of the T-cell transcriptome by hijacking the U2AF complex function. Through interaction with Tax, there is also a possibility of recruitment of the U2AF complex to regulate HTLV-1 pre-mRNAs splicing. In fact, the U2AF complex targets the 3' splice site of ~88% of protein coding transcripts [82], making it a perfect target for a transforming oncovirus.

# Material and methods

## Contact for reagents and resources

Further information and requests for resources and reagents (Table 1) should be directed to and will be fulfilled by the Lead Contact, Dr. Jean-Claude Twizere (email: jean-claude.twizere@uliege.be).

Plasmids and cell lines generated in this study are available upon request and approval of the Material Transfer Agreement (MTA) by the University of Liege.

## Methods and protocols

### Yeast strains

Yeast strains used in this study are derived from S288C as described [27]. Haploids of both mating type MATα Y8930C and MATa Y8800C were used. These strains harbor the following genotype: leu2-3,112 trp1-901 his3Δ200 ura3-52 gal4Δ gal80Δ GAL2::ADE2 GAL1:: HIS3@LYS2 GAL7::lacZ@MET2cyh2R. Yeast cells were cultured either in selective synthetic complete (SC) media or in rich medium (YPD) supplemented with glucose and adenine. Cells were incubated at 30˚C.

### Bacterial strains

Chemically competent DH5α E. coli cells were used for all bacterial transformation in this study. Post transformation, cells were cultured in LB (25 g/l) supplemented with antibiotics (50 μg/ml of ampicillin, spectinomycin or kanamycin) and incubated at 37˚C for 24 hours.

### Cell lines

HEK293T (*Homo sapiens*, fetal kidney) and HeLa (*Homo sapiens*, Henrietta Laks cervical cancer) cells were cultured in Dulbecco's Modified Eagle Medium (DMEM) supplemented with 10% fetal bovine serum, 2mmol/L L-glutamine and 100 I.U./mL penicillin and 100μg/mL streptomycin. Cells were incubated at 37˚C with 5% $CO_2$ and 95% humidity.

Jurkat (*Homo sapiens*, T-cell leukemia) cells were cultured in Roswell Park Memorial Institute (RPMI) media supplemented with 10% fetal bovine serum, 2mmol/L L-glutamine and 100 I.U./mL penicillin and 100 μg/mL streptomycin. Cells were incubated at 37˚C with 5% $CO_2$ and 95% humidity.

### Yeast two hybrid assay

Details of the Y2H screening method are described elsewhere [26,27]. Y2H assays were performed with 19 distinct fragments of the Tax protein and 9 fragments of HBZ protein cloned in CEN plasmids pDEST-AD-*CYH2* (AD vector) and pDEST-DB (DB vector). AD and DB vectors were transformed into MATa Y8800 and MATα Y8930 *Saccharomyces cerevisiae* strains, respectively [27]. A total of 3652 Human ORFs, corresponding to human transcription factors and RNA-binding proteins, were obtained from the human ORFeome v7.1 collection. For Y2H screening, we pooled MATa Y8800 yeast strains containing human and viral

**Table 1. Reagents and tools used in this study.** Reagents and tools are indicated with their source and references.

| REAGENT or RESOURCE | SOURCE | IDENTIFIER |
|---|---|---|
| Antibodies | | |
| Anti-SC35 | Abcam | PRID: [20,81] |
| Anti-Flag M2 | Sigma-Aldrich | PRID: AB_259529 |
| Anti-Flag (Rabbit) | Sigma-Aldrich | PRID: AB_439687 |
| Anti-Tax | Francoise Bex | Meertens et al., 2004 |
| Anti-GAPDH | Santa Cruz | PRID: AB_2107299 |
| Anti-HA | Santa Cruz | PRID: AB_627809 |
| Anti-myc | Santa Cruz | PRID: AB_2857941 |
| Chemicals | | |
| Doxycycline hydrochloride | Fisher Scientific | Cat#10592-13-9 |
| Protamine sulfate | MP Biomedicals | Cat#9009-65-8 |
| Blasticidin S | Sigma-Aldrich | Cat# 15205 |
| Hygromycin B | Sigma-Aldrich | Cat# 10843555001 |
| RNA-seq Raw and processed data | This paper | GEO: GSE146210 |
| Experimental Models: Cell Lines | | |
| HeLa | ATCC | CRM-CRL-2 |
| HEK293T | ATCC | CRL-3216 |
| Hek293 iHBZ | This paper | N/A |
| Jurkat | ATCC | CSC-C9455L |
| Lenti-X-293T cell | Clontech-Takara | Takara #632180 |
| Jurkat Tet3G iTax and Jurkat Tet3G iHBZ | This paper | N/A |
| Jurkat-iTax and Jurkat-iHBZ | This paper | N/A |
| Experimental Models: Organisms/Strains | | |
| *S.cerevisiae*:Strain background: MATa Y8800 | [27] | N/A |
| *S.cerevisiae*:Strain background: MATa Y8930 | [27] | N/A |
| Recombinant DNA | | |
| pLenti-CMVtight-Blast-DEST | Eric Campeau | Addgene#26434 |
| pLenti CMV rtTA3 | Eric Campeau | Addgene #26429 |
| pLVX-Tet3G | Clontech-Takara | Takara Cat#631187 |
| psPAX2 | Didier Trono | Addgene # 12260 |
| pCMV-VSVG | Didier Trono | Addgene #8454 |
| pLenti6 Tight-Tax | This paper | N/A |
| pLenti6 Tight-Flag-HBZ | This paper | N/A |
| pLV U6 shRNA hU2AF2 (mCherry-Neo) | This paper | N/A |
| humanORFeome v7.1 and v8.1 | The Center for Cancer Systems Biology (CCSB-DFCI) | https://ccsb.dana-farber.org/resources.html |
| pDEST-AD-CEN SY | [27] | N/A |
| pDEST-DB-CEN SY | [27] | N/A |
| pGLucN1 and pGLucN2 | [28] | N/A |
| Software and Algorithms | | |
| ImageJ | [84] | https://imagej.nih.gov/ij/ |
| STAR | [85] | https://code.google.com/archive/p/rna-star/ |
| DESeq2 | [86] | http://bioconductor.org/packages/release/bioc/html/DESeq2.html |
| rMATS | [46] | http://rnaseq-mats.sourceforge.net/ |
| GSEA 3.0 | [29] | https://software.broadinstitute.org/gsea/index.jsp |

*(Continued)*

**Table 1.** (Continued)

| REAGENT or RESOURCE | SOURCE | IDENTIFIER |
|---|---|---|
| Gorilla | [87] | http://cbl-gorilla.cs.technion.ac.il/ |
| AME | [57] | http://meme-suite.org/ |

AD-ORFs, and performed mating against MATα Y8930 yeast clones containing individual human or viral DB-ORFs. After selection of mated yeasts on medium lacking leucine, tryptophan and histidine, the identity of the interacting protein pairs was determined by sequencing of the corresponding ORF clone. A counter-selection on medium containing cycloheximide was performed simultaneously in the screens to identify false positives [27,88]. All discovered interacting pairs were retested again similarly but in pairwise screenings. Interacting pairs confirmed in the second screening were considered positives. Construction of interacting networks was done with Cytoscape v3.5.1 [89].

## Generation of Jurkat cell lines with inducible Tax or HBZ expression

The viral Tax gene and the FLAG-tagged HBZ gene were cloned into the pLenti-CMVtight-Blast-DEST (w762-1) vector from Addgene (#26434) with the Gateway cloning system. Production of lentiviral vectors (2nd generation) and generation of the Jurkat Tet3G iTax and Jurkat Tet3G iHBZ cell lines was performed by the GIGA Viral Vectors Platform at the University of Liège with the $3^{rd}$ generation inducible gene expression system Tet-On (from Clontech-Takara). The plasmids pLVX-Tet3G, pLenti6 Tight Tax and pLenti6 Tight Flag-HBZ were transfected separately on LentiX-293T Cell Line (from Clontech-Takara) each with a lentiviral packaging mix. This mix contained a packaging construct with the gag, pol, rev genes (psPAX2), and an Env plasmid expressing the vesicular stomatitis virus envelope glycoprotein G (VSV-G). The lentiviral supernatants were harvested, concentrated and tittered by qRT-PCR (Lentiviral Titration Kit LV900 from AbmGood) to be further used for transduction. The Jurkat cell lines ($5 \times 10^5$ cells/ml) were co-transduced with lentiviral vectors pLVX-Tet3G at a multiplicity of infection (MOI) of 30 and with pLenti6 Tight-Tax or -Flag-HBZ at a MOI of 16. For this transduction step, the reagent protamine sulfate (MP Biomedicals) was used according to the manufacturer´s instructions (8 μg/ml). Subsequently, the cells were centrifuged at 800 x g for 30 minutes at 37 degrees. The pellet was suspended in RPMI-1640 containing 10% FBS. After 72h cells were cultivated in cell culture medium containing blasticidin(10 μg/ml) and hygromycin B (400 μg/ml) in order to select transduced cells expressing the gene of interest (Tax or Flag-HBZ) and the rtTA3 gene until the non-transduced cells (negative control) died as determined by Trypan Blue staining.

## Generation of Jurkat shU2AF65 cells

Hek293 Lenti-x 1B4 cells (Clontech) were transfected with pLV U6 shRNA hU2AF2 (mCherry-Neo) (VectorBuilder), pVSV-G (Clontech, PT3343-5) and psPAX2 (Addgene, 12260). Supernatants were harvested and viruses were concentrated by ultracentrifugation. Jurkat cells were transduced with viruses at a MOI of 30. After 72h cells were selected with 2 mg/ml neomycin (G418, Invitrogen).

## Cell culture, transfections and treatments

Induction of Tax or Flag-HBZ in Jurkat-iTax or -iHBZ cells was performed by treating cells with doxycycline hydrochloride (1 μg/ml, Fisher Scientific) for 48h. HEK293T and HeLa cells were

transfected using polyethylenimine (PEI25K, Polysciences) (from 1 mg/ml) by a ratio 2:1 to plasmid concentration. HEK293T and HeLa cell lines were maintained in DMEM (Gibco) complemented with 10% FBS while Jurkat cell lines were maintained in RPMI (Gibco) also complemented with 10% FBS. Cell lines were regularly checked for mycoplasma contamination.

## RNA extraction, RT-PCR and RT-qPCR

Total RNA extraction from cell pellets was performed according to the manufacturer's protocol (Nucleo Spin RNA kit from Macherey-Nagel) and cDNA was obtained by reverse transcription with random primers using the RevertAid RT Reverse Transcription Kit from Thermo Fisher. One or half a microgram of total RNA was used to make cDNA, which were diluted 100 times to perform PCR amplification with specific primers, using Taq polymerase (Thermo Fisher). PCR products were migrated by SDS-PAGE and revealed with Gel Star (Lonza) under UV light. Bands were quantified using ImageJ software [84]. Validation of ASEs detected in Jurkat cells was performed by quantifying bands obtained after PCR amplification of regulated exons. The primers used target flanking exons and allowed detection of inclusion/exclusion of the alternatively spliced exon. Differential Percent Spliced In (ΔPSI) was calculated by subtracting the PSI value obtained from the Tax expressing condition (+Dox) to the PSI value from the CTRL condition (-Dox). PSI was calculated by dividing the quantification of inclusion and exclusion of the PCR band.

Quantitative PCR (qPCR) reactions were performed with iTaq Universal SYBR Green Supermix (BioRad) in triplicates on a LightCycler 480 instrument (Roche). The ΔΔCt method was used to analyze relative target mRNA levels with GAPDH as an internal control. All primers used in the study are presented in S4 Data.

## Immunofluorescence and confocal microscopy

Transfected HeLa cells with GFP-HBZ or an empty plasmid were grown on glass coverslips for 24h. Cells were washed in PBS and fixed with 4% paraformaldehyde (PAF) for 15 min. After washing with PBS, cells were permeabilized in PBS with 0.1% Triton X-100 for 5 min. Cells were then incubated in blocking solution (PBS with 4% BSA) for 1h before incubation with anti-SC35 primary antibody (Abcam) overnight at 4 degrees. Samples were then incubated with Alexa568-conjugated secondary antibodies (Thermo Fisher) and further incubated 10 min. with DAPI (Thermo Scientific, in PBS) before washing and mounting with Mowiol. All images were acquired with a Nikon A1 confocal microscope and processed with ImageJ.

## Transmission electron microscopy

Hek293 cells inducible for HBZ expression (Hek293 iHBZ) were generated similarly to Jurkat-iTax and -iHBZ cell lines. However, a plasmid pLenti CMV rtTA3 was used instead of the plasmid pLVX-Tet3G. HBZ expression was induced or not in Hek293 iHBZ cells for 48 hours with doxycycline before cells were fixed for 90 minutes at 4 degree C with 2.5% glutaraldehyde in a Sörensen 0.1 M phosphate buffer (pH 7.4) and post-fixed for 30 min with 2% osmium tetroxide. After dehydration in graded ethanol, samples were embedded in Epon. Ultrathin sections obtained with a Reichert Ultracut S ultra-microtome were contrasted with uranyl acetate and lead citrate. Observations were made with a Jeol JEM-1400 transmission electron microscope at 80kV.

## Cloning and plasmids

Viral clones were amplified by PCR using specific primers flanked at the 5' end with AttB1.1 and AttB2.1 Gateway sequences and were inserted into pDONR223 by Gateway cloning

(Invitrogen). Human ORFs (encoding U2AF65, SNRPA, eIF4A3) were retrieved from the human ORFeome v7.1 or the human ORFeome v8.1 (http://horfdb.dfci.harvard.edu). Inserts in pDONR223 were then transferred into appropriate destination vectors by LR cloning. A list of Tax and HBZ clones is available in S4 Data.

### Cell lysates, immunoprecipitation, immunoblotting and antibodies

For Immunoprecipitations (IPs) HEK293T cells were harvested and lysed in IPLS (Immuno-precipitation Low Salt; Tris-HCl pH 7.5 50 mM, EDTA, pH 8, 0.5 mM, 0.5% NP-40, 10% glyc-erol, 120 mM NaCl, complete Protease Inhibitor (Roche)). RNaseA treatment (Thermo Fisher Scientific, 10 $\mu$g/ml at 37% for 30 min) was performed on cleared lysates when indicated. Supernatants were incubated with anti-FLAG M2 agarose beads (Sigma-Aldrich) and then were washed with IPLS (incubation times and number of washes depended on tested interac-tions). For semi-endogenous IPs, rabbit anti-Tax antibody [90] was incubated overnight with cell lysates. Afterwards, Protein A/G PLUS-Agarose beads (Santa Cruz) were incubated with lysates for 2h and beads were washed 3 times with IPLS. Beads were re-suspended in 2x SDS loading buffer and boiled. Samples were then analyzed by SDS-PAGE and western blotting and revealed with ECL detection kit (GE Healthcare Bio-Sciences) according to standard procedures.

### Protein Complementation Assay (PCA)

Tax, HBZ and interacting partners ORFs were cloned in destination vectors containing GLucN1 and GLucN2 fragments of the *Gaussia princeps* luciferase. HEK293T cells were seeded in 24-well or 96-well plates and transfected with 500 ng or 200 ng of the appropriate constructs (GLucN1 + GLucN2), respectively. After 24h cells were washed with PBS and lysed using the manufacturer's lysing buffer (Renilla luciferase kit, Promega). Ten to 20 $\mu$l of lysates were then used to quantify luminescence in a Centro lb 960 luminometer (Berthold).

### RNA-seq data analysis

Libraries were prepared with the Illumina Truseq stranded mRNA sample prep kit and paired-end sequencing was performed with the Illumina NextSeq500 PE2X75 system by the Geno-mics platform at GIGA, University of Liege. Three replicates were made for each condition analysed (Jurkat iTax +Dox, Jurkat iTax -Dox, Jurkat iHBZ +Dox and Jurkat iHBZ -Dox). Sequence reads were aligned to the human genome hg19 (UCSC) using STAR [85]. Differen-tial expression analysis was performed with DESeq2 [86] on read counts from STAR quant Mode. Genes were considered significantly up- or down-regulated if their base 2 logarithm fold change was >1 or <-1 and their adjusted p-value was <0.01. Analysis of Alternative Splic-ing Events (ASEs) was performed with the rMATS software (v3.2.1 for the Japanese cohort and iTax iHBZ and v4.1.1 for the Afro-Caribbean cohort), using reads mapped with STAR that spanned between exon junctions of ASEs [46]. rMATS outputs reported differential ASEs by calculating the difference in PSI between two conditions, namely ΔPSI, ranging from -1 to +1. For instance, SE events where exons were more excluded presented a ΔPSI value <0 while exons that were more included had a ΔPSI >0. Genes with low expression levels were removed from the output results by filtering out genes with a TPM <1. TPM was calculated using Salmon (v0.9.1 for the Japanese cohort, iTax and iHBZ, and v1.4.0 for the Afro-Caribbean cohort). ASEs were further filtered to consider only events with a ΔPSI >0.1 or <-0.1 and with a FDR <0.05. Sashimi plots were generated using rmats2sashimi.

RNA-seq data from the Japanese patient cohort were previously described (Kataoka et al., 2015). For the purpose of this study, we selected: 35 ATLL patient samples (17 acute, 11

chronic, 4 lymphoma, 2 smoldering, 1 unknown), 3 samples from asymptomatic individuals infected with HTLV-1 (ACs) and CD4+ T-cells from 3 healthy volunteers. RNA-seq data from the Afro-Caribbean cohort were previously described [8,55,56]. We selected 29 ATLL cases (all of the acute subtype), 5 HTLV-1 infected asymptomatic carriers (ACs) and 4 CD4+ T-cell samples from healthy volunteers.

## Gene enrichment analysis

Gene enrichment analysis on differentially expressed genes was performed with GSEA 3.0 (Gene Set Enrichment Analysis) in pre-ranked mode using the hallmark gene sets from the Molecular Signatures Database [29]. GO enrichment analysis was performed on alternatively spliced genes with Gorilla [87] using all detected ASEs as a background. Other enrichment analyses were performed by hypergeometric tests or by calculating empirical p-values in R.

## RNA-Binding motif enrichment analysis

Sequences of regulated exons detected by rMATS (SE events only) and up to 200 bp of their flanking introns were retrieved using bed tools v0.9.1. Control sequences consisted of detected ASEs with low ΔPSI values and high FDR. The FASTA files generated were then interrogated for known RNA-binding motifs from the literature [91] using the AME software from MEME suite 5.0.2 with default settings [57]. Motifs with an adjusted p-value <0.05 were considered significant.

## Supporting information

**S1 Fig. Overview of the Yeast-two hybrid (Y2H) experiment. (A)** Schematic representation of positive interaction between Tax, CREB/ATF and CBP/P300 on the viral promoter. **(B)** As in (A) but negative interaction driven by HBZ. **(C)** Illustrative network diagram showing interactions between Tax and HBZ with cellular transcription factors (TF) or RNA-binding proteins (RBP). **(D)**. Pipeline to generate a comprehensive list of human RBPs and TFs. **(E)** Cloning strategy for the Tax/HBZ mini-library. **(F)** Y2H strategy to identify high-quality protein-protein interactions (PPIs).
(TIF)

**S2 Fig. Tax and HBZ mutants used for the Yeast-two hybrid (Y2H) experiment.** Representation of the functional and structural domains of (A) Tax and (B) HBZ. (A) Tax deletion and point mutants used for Y2H screening are described below Tax diagram, with references as PMIDs. (B) HBZ deletion mutants used for Y2H screening are depicted below the diagram. NLS = nuclear localization signal, NES = nuclear export signal, LZR = leucine zipper-like motif regions, PBM = PDZ domain binding motif.
(TIF)

**S3 Fig. A comprehensive map of host proteins interacting with Tax and HBZ.** Host proteins are color-coded according to their function in Transcriptional Regulation (TR), Post-Transcriptional Regulation (PTR) or other (purple). Upward triangles and downward arrows show genes with an up-regulation or down-regulation following Tax or HBZ expression, respectively.
(TIF)

**S4 Fig. Comparative expression of Tax and HBZ in Jurkat cells and in primary cells, and perturbation of nuclear speckles and interchromatin granules by Tax and HBZ. (A-B)** Log2TPM of *Tax*, *HPRT1*, *GAPDH* and *HBZ* mRNA expression in Jurkat-iTax (A) or Jurkat-i-

HBZ (B). (**C-D**) Normalized Tax and HBZ mRNA expression in ATL, TSP and asymptomatic carrier (AC) samples. qPCR data are normalized to the expression of HPRT1 mRNA. (**E-F**) qRT-PCR analysis showing variation of normalized mRNA abundance of (**E**) *SYT4A* and (**F**) *STAT5A* upon expression of Tax and HBZ. * = p-value <0.05, ** = p-value <0.01, *** = p-value <0.001. (**G**) Immunofluorescence microscopy indicates that SC35, a marker of nuclear speckles, localizes into more round shapes upon expression of HBZ in HeLa cells. Scale bars = 2 $\mu$m. (**H**) Representative nuclear portion of HEK293T expressing HBZ(left) or not (right), visualized by TEM. Clusters of interchromatin granules (also known as nuclear speckles) are surrounded by red dotted lines and display a more compact phenotype upon HBZ expression. C = condensed chromatin. Scale bars = 1 $\mu$m.
(TIF)

**S5 Fig. Alternatively spliced exons shared between ATLL patient samples from the Japanese cohort, Jurkat-iTax and Jurkat-iHBZ cells.**
(TIF)

**S6 Fig. Sashimi plots of significant ASEs affecting PTPRC exons 4 to 7 in the Japanese cohort.** Splicing events are depicted above each sashimi plot, coordinates of regulated and flanking exons are indicated at the bottom. (A) Increased inclusion of exon 4. Decreased inclusion of exon 5 (B), 6 (C) and 7 (D). For ATLL, sashimi plot is a representative case.
(TIF)

**S7 Fig. Spatial distribution of U2AF2 binding-motif in ATLL patients of the Japanese cohort, No-interaction between HBZ and the U2AF complex, and number of T nucleotides in 3'SS of SE in Jurkat-iHBZ cells. (A)** Solid lines indicate the mean U2AF2 binding motif score calculated in a 50 bp sliding window. Dotted lines indicate -log10 p-values obtained by statistical comparison of motif scores between modified exons (exclusion = down-regulated and inclusion = up-regulated) against non-modified background exons. Green box represents regulated exons flanked by neighboring introns and upstream and downstream exons in black lines and grey boxes. (**B**) GPCA to test the absence of interaction between HBZ and U2AF complex subunits (U2AF65 and U2AF35). Y-axis shows luciferase activity for a representative experiment of 3 repetitions. (**C**) Number of Ts in the 30bp upstream of 3'SS of alternatively spliced exons (SE events) in Jurkat-iHBZ cells. Medians are represented by a line. Included P <2.2e-16, Excluded P = 0.03398, by Welch Two Sample t-test in R.
(TIF)

**S1 Data. Interactome and transcriptome data.** Data sheet 1: Tax interactome data from this work and previous studies. Data sheet 2: HBZ interactome data from this work and previous studies. Data sheet 3: Results of the Y2H screen for Tax and mutant constructs. Data sheet 4: Results of the Y2H screen for HBZ and individual domain constructs. Data sheet 5: RNA-binding motifs identified using AME showing RBPs with enriched motifs in Skipped Exons genes from ATLL samples. Data sheet 6: Shared differential expressed genes between ATLL, JurkatiTax JurkatiHBZ; and HTLV-1 asymptomatic carriers.
(XLSX)

**S2 Data. Transcriptome data.** Data sheet 1: Transcriptome DESeq results of JurkatiTax samples analyses. Data sheet 2: Transcriptome DESeq results of JurkatiHBZ samples analyses. Data sheet 3: GSEA results of JurkatiTax samples analyses. Data sheet 4: GSEA results of JurkatiHBZ samples analyses. Data sheet 5: Shared DEGs between JurkatiTax and JurkatiHBZ. Data sheet 6: Shared DEGs between ATLL, JurkatiTax and JurkatiHBZ; but not found in HTLV-1 asymptomatic carriers. Data sheet 7: Shared DEGs regulated in opposite direction between

JurkatiTax and JurkatiHBZ.
(XLSX)

**S3 Data. Alternative splicing events (ASEs) data.** Data sheet 1: ASEs detected in Jurkat iTax. Data sheet 2: ASEs detected in Jurkat iHBZ. Data sheet 3: Shared events Jurkat iTax, Jurkat iHBZ. Data sheet 4: GOrilla results AS genes in Jurkat iTax. Data sheet 5: GOrilla results AS genes in Jurkat iHBZ. Data sheet 6: Known cancer genes presenting AS in Jurkat iTax. Data sheet 7: Known cancer genes presenting AS in Jurkat iHBZ. Data sheet 8: Genes presenting DEGs and ASEs. Data sheet 9: ASEs detected in HTLV-1 carriers of the Japanese cohort. Data sheet 10: ASEs detected in ATLL of the Japanese cohort. Data sheet 11: Shared events Jurkat iTax, ATLL of the Japanese cohort. Data sheet 12: Shared events Jurkat iHBZ, ATLL of the Japanese cohort. Data sheet 13: Shared genes with ASEs between ATLL, Jurkat iTax, Jurkat iHBZ but not in HTLV-1 carriers of the Japanese cohort. Data sheet 14: ASEs detected in HTLV-1 carriers of the Afro-Caribbean cohort. Data sheet 15: ASEs detected in ATLL of the Afro-Caribbean cohort. Data sheet 16: Shared events Jurkat iTax, ATLL of the Afro-Caribbean cohort. Data sheet 17: Shared events Jurkat iHBZ, ATLL of the Afro-Caribbean cohort. Data sheet 18: Genes presenting ASEs in both cohorts. Data sheet 19: PPIs in the 26 and 50 shared genes.
(XLSX)

**S4 Data. The lists of Tax and HBZ constructs, and primers used in this work.** Data sheet 1: A list of Tax and HBZ cDNA clones. Data sheet 2: A list of primers used in RT-PCR experiments.
(XLSX)

## Acknowledgments

We thank Dr. Clara L. Kielkopf (University of Rochester Medical Center, USA) and Dr. Yves Jacob (Institut Pasteur, Paris, France) for DNA constructs. We thank the following University of Liege core facilities: Cell Imaging, Genomics and Viral vectors for their services. We thank Patricia Piscicelli for technical assistance in TEM. In memory of Prof. Renaud Mahieux who made a profound impact on the advancement of molecular understanding of the roles of Tax and HBZ. He was a member of the evaluation committee for C.V's. PhD thesis behind this work.

## Author Contributions

**Conceptualization:** Franck Dequiedt, Jean-Claude Twizere.

**Data curation:** Charlotte Vandermeulen, Georges Coppin, Keisuke Kataoka, Seishi Ogawa, Johan Van Weyenbergh.

**Formal analysis:** Charlotte Vandermeulen, Tina O'Grady, Jerome Wayet, Georges Coppin, Lamya Ben Ameur, Marc Thiry, Franck Mortreux, Johan Van Weyenbergh, Jean-Marie Peloponese, Benoit Charloteaux.

**Funding acquisition:** Johan Van Weyenbergh, David E. Hill, Marc Vidal, Franck Dequiedt, Jean-Claude Twizere.

**Investigation:** Charlotte Vandermeulen, Jerome Wayet, Bartimee Galvan, Sibusiso Maseko, Marc Thiry, Jean-Marie Peloponese, Benoit Charloteaux, Anne Van den Broeke, Franck Dequiedt, Jean-Claude Twizere.

**Methodology:** Charlotte Vandermeulen, Tina O'Grady, Jerome Wayet, Bartimee Galvan, Sibusiso Maseko, Majid Cherkaoui, Alice Desbuleux, Lamya Ben Ameur, Marc Thiry, Franck Mortreux, Johan Van Weyenbergh, Jean-Marie Peloponese, Benoit Charloteaux, Anne Van den Broeke, Jean-Claude Twizere.

**Project administration:** Jean-Claude Twizere.

**Resources:** Keisuke Kataoka, Seishi Ogawa, Olivier Hermine, Ambroise Marcais, Franck Mortreux, Johan Van Weyenbergh.

**Software:** Charlotte Vandermeulen, Tina O'Grady, Benoit Charloteaux.

**Supervision:** Franck Mortreux, Michael A. Calderwood, Benoit Charloteaux, David E. Hill, Marc Vidal, Franck Dequiedt, Jean-Claude Twizere.

**Validation:** Charlotte Vandermeulen, Jean-Claude Twizere.

**Visualization:** Julien Olivet, Marc Thiry.

**Writing – original draft:** Charlotte Vandermeulen, Franck Dequiedt, Jean-Claude Twizere.

**Writing – review & editing:** Charlotte Vandermeulen, Tina O'Grady, Julien Olivet, Benoit Charloteaux, Anne Van den Broeke, David E. Hill, Marc Vidal, Franck Dequiedt, Jean-Claude Twizere.

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
