## [Decision Letter · Decision Letter 0]

23 Apr 2021

Dear Professor TWIZERE,

Thank you very much for submitting your manuscript "The viral oncoproteins Tax and HBZ reprogram the cellular mRNA splicing landscape" for consideration at PLOS Pathogens. As with all papers reviewed by the journal, your manuscript was reviewed by members of the editorial board and by several independent reviewers. In light of the reviews (below this email), we would like to invite the resubmission of a significantly-revised version that takes into account the reviewers' comments.

All three reviewers found your study of interest, novelty and potential importance. However, all three also raised significant issues that need to be addressed. The most important points concern 1. the relationship between Jurkat cells and primary T cells, 2. the difference (if any) in the level of expression of Tax and HBZ between the inducible Jurkat cell lines and primary cells (whether non-malignant or ATL cells), and 3. the need to balance the numbers of control samples with the experimental samples. In addition, the reviewers asked for more details on the methods used, in particular the bioinformatic analysis.

We cannot make any decision about publication until we have seen the revised manuscript and your response to the reviewers' comments. Your revised manuscript is also likely to be sent to reviewers for further evaluation.

Sincerely,

Charles R. M. Bangham

Associate Editor

PLOS Pathogens

Susan Ross

Section Editor

PLOS Pathogens

Kasturi Haldar

Editor-in-Chief

PLOS Pathogens

orcid.org/0000-0001-5065-158X

Michael Malim

Editor-in-Chief

PLOS Pathogens

orcid.org/0000-0002-7699-2064

All three reviewers found your study of interest, novelty and potential importance. However, all three also raised significant issues that need to be addressed. The most important points concern 1. the relationship between Jurkat cells and primary T cells, 2. the difference (if any) in the level of expression of Tax and HBZ between the inducible Jurkat cell lines and primary cells (whether non-malignant or ATL cells), and 3. the need to balance the numbers of control samples with the experimental samples. In addition, the reviewers asked for more details on the methods used, in particular the bioinformatics.

Reviewer's Responses to Questions

**Part I - Summary**

Reviewer #1: The authors have screened interacting host factors binding with HTLV-1 Tax and HBZ by using Y2H system. They identified distinct but also common host factors between Tax and HBZ. They further performed comparative analysis of transcriptomic changes associated with Tax and HBZ using Jurkat with HBZ- and Tax-inducible cells. They found Tax and HBZ has an impact on cellular gene alternative splicing events. The aberrant alternative splicing events were found in ATLL patients. Tax and HBZ interact with RNA processing factors, such as U2AF2, and thereby inducing alternative splicing events.

The findings shown are potentially interest contain some novelty. However, some figures are difficult to understand and also there is limited description about the bioinformatic method used. Some data presented seems to be preliminary and not so convincing. It would be better to add more explanation and evidence to support authors’ proposed title “The viral oncoproteins Tax and HBZ reprogram the cellular mRNA splicing landscape”.

Reviewer #2: In the present study, Vandermeulen investigated the effects of Tax and HBZ on the transcriptome and interactome in Jurkat cells. Results uncovered many interactions with RNA-binding proteins, including U2AF2, a key cellular regulator of pre-mRNA splicing. Consistent with this finding, Tax and HBZ perturbed the splicing landscape of Jurkat cells. Interestingly, Tax favored exon inclusion while HBZ induced exon exclusion. A subset of Tax- and HBZ-dependent splicing changes were also found in cells from ATLL patients. The study is potentially interesting and investigates the effects of 2 key HTLV-1 proteins (Tax and HBZ) on a poorly explored field – i.e. the regulation of the host RNA splicing by viral factors.

Reviewer #3: Vandermeulen et al. in their manuscript entitled “The viral oncoproteins Tax and HBZ reprogram the cellular mRNA splicing landscape” analyzed the transcriptome and interactome of the HBZ and Tax viral proteins. The authors found that each protein has distinct targets, but also share common transcription factors and RNA-binding proteins. By analyzing the cellular splicing landscape using rMATS software, HBZ and Tax were found to alter the splicing landscape in T-cells. Interestingly, they exert opposing effects with Tax inducing exon inclusion and HBZ inducing exon exclusion. Several of the splicing changes identified by the authors were also altered in ATL patients. The manuscript was well-written and is of high importance to both the HTLV-1 field and other oncogenic or pathogenic viral fields. However, several comments/suggestions are provided below for the authors’ consideration in order to satisfy the PLOS Pathogens criteria for publication.

**Part II – Major Issues: Key Experiments Required for Acceptance**

Reviewer #1: 1. Alternative splicing analysis. Fig. 3A

It is not well explained in current manuscript about how they define alternative splicing events. In the method, “ASEs were filtered to consider only events with a ΔPSI >1 or <-1 and with a FDR <0.05”. Just a few percent variation of PSI can be caused by experimental replication. The authors need to explain more about data representativeness. How many replicates they performed for RNA-seq analysis, for example (Jurkat and patient samples). It is difficult to understand Fig. 3E. How did they make this graph? In general, RT-PCR may not be good to distinguish small change of cDNA amount. ddPCR would be useful for the purpose. In Fig. 7 L-O, there were only less than 1 percent difference in PSI by RT-PCR. It is hard to say there is differences. I may misunderstand the graph, so please explain more in details.

2. Figure 4

RNA transcription and splicing are basically distinct biological processes, even though there is some relationship between them. Thus, we can expect the observation that only a small number of genes (23 and 29 for Tax or HBZ, respectively) presenting ASEs were differentially expressed (Fig. 3 C and D). It is not clear for me that the results is valuable to show as figure.

3. Figure 5

The result of infected individuals is interesting and important. Please describe the details of RNA-seq.

A. Are they generated using total PBMCs or CD4+ T-cell population? If they are from PBMCs, different proportion of PBMC subset would affect the result. If they are from CD4 T cells, there is still different proportion of CD4 T cell subset between heathy donor and HTLV-1-infected individuals. Effector/memory CD4 T cell fraction is known to be increase in HTLV-1-infected individuals. The author had better mention the possibility.

B. How about the expression level of tax and HBZ in RNA-seq data? That would be informative to analyze relationship between Tax/HBZ and ASEs.

Reviewer #2: Major comments

Jurkat cells have an early T-cell phenotype and thus might not represent a good cell model to study HTLV-1 which infects and persists in mature T-cells. How do the levels of the proteins in the Jurkat system compare to the levels observed in productively infected cells or ATLL cells?

The validation in patient samples is of key importance to assess the relevance of the results presented; however the cohorts examined are poorly matched in terms of numbers (35 ATLL, 3 HC, 3 normal controls), which is likely to skew the results. Furthermore there is no indication about the % of infected cells in the ATLL and HC samples. The comparison should be done between samples with similar % of infected cells, otherwise infected cells should be sorted.

Reviewer #3: 1. The authors describe using ‘a mini-library of different Tax and HBZ clones with previously described functional domains’ on page 9. A brief description of the domains used in this study would be beneficial for the field. Also, was there sequence homology between the different clones for both Tax and HBZ?

2. This study utilized an inducible cellular system to express Tax or HBZ, separately. What is the expression of inducible-Tax or -HBZ relative to naturally infected cells? Are the effects observed through the interactome and transcriptome results long-lasting or transient (what would happen if you took away Tax or HBZ)?

3. In these experiments, was the hbz mRNA also present? Hbz mRNA has been shown to have proliferative effects. It’s difficult to say whether the transcriptome results found were solely due to the protein, the mRNA or both.

**Part III – Minor Issues: Editorial and Data Presentation Modifications**

Reviewer #1: 0. There are several previous papers about HTLV-1 and RNA splicing(PMID 25519886, PMID 3254671, PMID 23600753). The authors should use them as references and discuss current study result by considering previous reports.

1. Fig. 1A-C can be shown as supplementary figures.

2. Fig. 1E: What Overlap and Union means?

3. Fig. 1G: There is no description in the paper. If so, please delete that.

4. Fig. 2E-F: Please check again if “P=0.00” or “q = 0.00” is correct or not.

5. Fig. 3E: Please provide more information how they analyzed and showed the figure.

6. Fig. 6: It is not clear how to see this figure. There seems to be little graphical difference. Is the difference is statistically significant?

7. Fig. 7C: Please explain more in details. Which molecules are associated with Tax or HBZ?

8. Fig. 7E: The result is not convincing. I suggest to revised the figure.

9. Table S2: 12 genes whose expression was altered in opposite directions between Tax and HBZ are interesting. It would be useful to show the list.

Reviewer #2: Additional comments

There are no page and line numbers

The title of the manuscript should specify ‘HTLV-1’

The first sentence of the abstract starts with ‘while’. This is poor English.

In the Synopsis, the first statement should specify ‘HTLV-1-encoded proteins Tax and HBZ’ and not the second.

First paragraph of the Introduction: Hayward et al., 1981 is not in the reference list.

Second paragraph of the Introduction: The review by Matsuoka and Mesnard on HBZ (Retrovirology 2020) and by Mohanty and Harhaj on Tax (Pathogens 2020) would provide more up to date information compared to the cited reviews.

Results - First section:

Figures 1A, 1B and 1C are very simplistic and could be removed or placed in the supplemental section. Some of the Tax mutants used in the yeast experiments predate the references provided in the text. For example, M10, M22 and M47 were described by Smith MR and Greene WC (Genes Dev 1990). Table S1 should provide a brief description of these and other mutants whose names are not self-explanatory. It would also be helpful to include a diagram of the domain structures of Tax and HBZ so that the reader understands the locations of the Tax mutations and HBZ domains tested. Figure 1E presents information that is not important. It could be removed or moved to the Supplemental section. Figure S1 lacks a description of panels D and E. How many repeats were performed? What do the error bars represent? The paragraph describing Figure 1F highlights that Tax and HBZ have many more PPIs compared to known cancer gene products or other tumor virus proteins, presumably based on 2 cited references from 2014 and 1012, respectively. It is likely that more interactions have been described in the intervening years, so the relevance of this comparison is questionable.

Results - Second section:

The statement that Tax and HBZ are differentially expressed in vivo is not accurate, in particular if one considers infected cells, and not ATLL cells. The is no need to use this argument to justify the use of Jurkat cells expressing the individual proteins.

In Figure S3A, the levels of the housekeeper protein HSP70 are very different. When this difference is compared to the levels of Tax and HBZ detected, the difference in the levels of these proteins becomes enormous. How do the levels of the proteins in the Jurkat system compare to the levels observed in productively infected cells or ATLL cells?

According to the Methods section, the confocal analysis was carried out in HeLa cells but the legend to Figure S3D refers to HEK293T cells. This needs to be corrected. The images shown in Panel E are not convincing. In general, it would have been preferable to carry out these experiments in a cell system that better approximates mature T-cells. Also, the title of this figure does not describe Panels A-C. In Panels B and C ‘Normalized’, not ‘Noralized’. What do the asterisks mean?

Results - Third section:

The first sentence talks about transcriptional ‘defects’. This should read ‘effects’. The information shown in Figure 3A is not essential. Figure 3E is cryptic. It would be more helpful to show the results of the RT-PCR assays of the individual genes as bar graphs.

Results - Fourth Section:

Figure 4B should list the genes in alphabetical order. The 2 panels in Figure 4C should labeled ‘Tax’ and ‘HBZ’ to help the reader. The title to Figure S4 should be the same as the one that appears in the figure. Are the ASes listed in the figure shared among the 3 types of samples? As written, this is unclear. The title to Figure S5 does not describe panels B and C. What do the lines in the bars represent? Means or Medians? They look identical for unchanged vs excluded. The p values don’t seem to correspond to the differences as visualized in the figure.

Results - Fifth Seciton:

The pie chart in Figure 5A might have a problem - the section representing MXE (19.5%) seems to cover about 1/3 of the pie.

Results - Sixth section:

Figure 7C indicates HuR. The proper gene symbol is ELAVL1. Official gene symbols should be used in the figures; common names can be included in the text.

Reviewer #3: 1. The ‘Synposis’ section was somewhat confusing without a figure legend describing the image.

2. Figure 1B: HBZ has been shown to interact with CREB and prevent it from binding to the viral LTR CRE binding sites (i.e. sequesters CREB away from promoter). The diagram shown here suggests the complex is bound at the chromatin and inhibits transcription.

3. In Figure 7E, the authors should distinguish the background band (upper band?) and Tax band in the Tax western blot. Also, in what cells were these experiments performed?

4. Could Tax or HBZ have any effect on viral RNA splicing?

PLOS authors have the option to publish the peer review history of their article (what does this mean?). If published, this will include your full peer review and any attached files.

Reviewer #1: No

Reviewer #2: No

Reviewer #3: No
---

## [Decision Letter · Decision Letter 1]

19 Aug 2021

Dear Professor TWIZERE,

Thank you very much for submitting your manuscript "The HTLV-1 viral oncoproteins Tax and HBZ reprogram the cellular mRNA splicing landscape" for consideration at PLOS Pathogens. As with all papers reviewed by the journal, your manuscript was reviewed by members of the editorial board and by several independent reviewers. The reviewers appreciated the attention to an important topic. Based on the reviews, we are likely to accept this manuscript for publication, providing that you modify the manuscript according to the review recommendations.

While two reviewers are content with your response to the points raised, I agree with Reviewer 2 that severe limitations remain in the reliance on Jurkat cells to infer normal physiological actions of the viral proteins. I would therefore ask you to clearly and explicitly acknowledge the limitations of this approach in the Discussion section; the current text at lines 9-12 on p.14 refer to the interplay between the viral proteins, rather than to the nature of the cell line or the potentially unphysiological level of expression of the proteins. Second, lines 4-6 on p.14 are potentially misleading: it is not clear whether the phrase 'in the initial steps of HTLV-1 infection' concerns only ATLL cells or non-malignant infected cells as well, and whether it refers to primary infection of the host or to new infection of a single cell. Please clarify this sentence.

Sincerely,

Charles R. M. Bangham

Associate Editor

PLOS Pathogens

Susan Ross

Section Editor

PLOS Pathogens

Kasturi Haldar

Editor-in-Chief

PLOS Pathogens

orcid.org/0000-0001-5065-158X

Michael Malim

Editor-in-Chief

PLOS Pathogens

orcid.org/0000-0002-7699-2064

While two reviewers are content with your response to the points raised, I agree with Reviewer 2 that severe limitations remain in the reliance on Jurkat cells to infer normal physiological actions of the viral proteins. I would therefore ask you to clearly and explicitly acknowledge the limitations of this approach in the Discussion section; the current text at lines 9-12 on p.14 refer to the interplay between the viral proteins, rather than to the nature of the cell line or the potentially unphysiological level of expression of the proteins. Second, lines 4-6 on p.14 are potentially misleading: it is not clear whether the phrase 'in the initial steps of HTLV-1 infection' concerns only ATLL cells or non-malignant infected cells as well, and whether it refers to primary infection of the host or to new infection of a single cell. Please clarify this sentence.

Reviewer Comments (if any, and for reference):

Reviewer's Responses to Questions

**Part I - Summary**

Reviewer #1: The authors have screened interacting host factors binding with HTLV-1 Tax and HBZ by using Y2H system. They identified distinct but also common host factors between Tax and HBZ. They further performed comparative analysis of transcriptomic changes associated with Tax and HBZ using Jurkat with HBZ- and Tax-inducible cells. They found Tax and HBZ has an impact on cellular gene alternative splicing events. The aberrant alternative splicing events were found in ATLL patients. Tax and HBZ interact with RNA processing factors, such as U2AF2, and thereby inducing alternative splicing events.

The findings shown are interest and will add several key mechanisms underlying HTLV-1 infection.

Reviewer #2: In their revised submission Vandermuele et al. have made an appreciable effort to address some of the points raised by the reviewers and the quality of the manuscript has improved accordingly.

However, some of the main concerns that I had raised on the original submission still remain (see below).

Reviewer #3: Vandermeulen et al. in their manuscript entitled “The viral oncoproteins Tax and HBZ reprogram the cellular mRNA splicing landscape” analyzed the transcriptome and interactome of the HBZ and Tax viral proteins. The authors found that each protein has distinct targets, but also share common transcription factors and RNA-binding proteins. By analyzing the cellular splicing landscape using rMATS software, HBZ and Tax were found to alter the splicing landscape in T-cells. Interestingly, they exert opposing effects with Tax inducing exon inclusion and HBZ inducing exon exclusion. Several of the splicing changes identified by the authors were also altered in ATL patients. The manuscript was well-written and is of high importance to both the HTLV-1 field and other oncogenic or pathogenic viral fields. The authors have successfully addressed most of the concerns of the reviewers and now satisfy the PLOS Pathogens criteria for publication.

**Part II – Major Issues: Key Experiments Required for Acceptance**

Reviewer #1: The authors have answered reviewers' questions by adding new analysis.

Reviewer #2: Main points:

1 - The Authors present a legthy justification for using jurkat cells as a model of HTLV-1 infection/persistence, however this cell line remains suboptimal for HTLV-1 studies.

2 - The expression levels of Tax and HBZ measured by the Authors in Jurkat cells differ greatly from those estimated in ATL cells (137-fold higher for Tax and 50-fold lower for HBZ), suggesting that the effects observed in the jurkat model might not faithfully represent what happens in patients. The Authors' justification that the intra-group variability in patients is also high is not a strong argument.

3 - The validation studies in patients are still weakened by the poor match between controls and patients. Instead of testing a larger number of controls, the Authors analyzed in silico data from patients of different ethnicity/geographical origin. This is a rather convoluted and indirect approach. It would have been much better to actually test an appropriate number of controls.

Reviewer #3: None

**Part III – Minor Issues: Editorial and Data Presentation Modifications**

Reviewer #1: The authors have answered reviewers' questions by revising manuscripts.

Reviewer #2: n.a.

Reviewer #3: None

PLOS authors have the option to publish the peer review history of their article (what does this mean?). If published, this will include your full peer review and any attached files.

Reviewer #1: No

Reviewer #2: No

Reviewer #3: No

Figure Files:

Data Requirements:

Reproducibility:

References:

---

## [Editor Report · Decision Letter 2]

27 Aug 2021

Dear Professor TWIZERE,

We are pleased to inform you that your manuscript 'The HTLV-1 viral oncoproteins Tax and HBZ reprogram the cellular mRNA splicing landscape' has been provisionally accepted for publication in PLOS Pathogens.

Best regards,

Charles R. M. Bangham

Associate Editor

PLOS Pathogens

Susan Ross

Section Editor

PLOS Pathogens

Kasturi Haldar

Editor-in-Chief

PLOS Pathogens

orcid.org/0000-0001-5065-158X

Michael Malim

Editor-in-Chief

PLOS Pathogens

orcid.org/0000-0002-7699-2064
---

## [Editor Report · Acceptance letter]

14 Sep 2021

Dear Professor TWIZERE,

We are delighted to inform you that your manuscript, "The HTLV-1 viral oncoproteins Tax and HBZ reprogram the cellular mRNA splicing landscape," has been formally accepted for publication in PLOS Pathogens.

Best regards,

Kasturi Haldar

Editor-in-Chief

PLOS Pathogens

orcid.org/0000-0001-5065-158X

Michael Malim

Editor-in-Chief

PLOS Pathogens

orcid.org/0000-0002-7699-2064